# Hunt Instead of Wait: Evaluating Deep Data Research on Large Language Models

Wei Liu [1]   Peijie Yu [2]   Michele Orini [1]   Yali Du [1 3]   Yulan He [1 3]

## Abstract

The agency expected of Agentic Large Language Models goes beyond answering correctly, requiring autonomy to set goals and decide what to explore. We term this *investigatory intelligence*, distinguishing it from *executional intelligence*, which merely completes assigned tasks. Data Science provides a natural testbed, as real-world analysis starts from raw data rather than explicit queries, yet few benchmarks focus on it. To address this, we introduce **Deep Data Research (DDR)**, an open-ended task where LLMs autonomously extract key insights from databases, and **DDR-Bench**, a large-scale, checklist-based benchmark that enables verifiable evaluation. Results show that while frontier models display emerging agency, long-horizon exploration remains challenging. Our analysis highlights that effective investigatory intelligence depends not only on agent scaffolding or merely scaling, but also on intrinsic strategies of agentic models.

## 1. Introduction

Agentic large language models (Agentic LLMs) extend conventional LLMs (Brown et al., 2020; OpenAI, 2022; 2023; 2025) from reactive response generation to interactive decision making (Zhang et al., 2025b). Recent agentic systems have demonstrated the ability to complete complex real-world tasks that require long-horizon behaviour (Plaat et al., 2025; Wang et al., 2025a; Yao et al., 2023; Xu et al., 2024; Wang et al., 2025b; Yehudai et al., 2025). However, most existing evaluations implicitly assume that the goal is specified in advance, and primarily assess models on their ability to execute predefined objectives. This framing conflates two distinct forms of intelligence: *executional intelligence*,

which concerns how well an agent carries out a given task, and *investigatory intelligence*, which concerns whether the agent can autonomously decide what is worth investigating. Current evaluation frameworks rarely assess it.

*AI for Data Science* provides a natural setting to study investigatory intelligence. Human analysts typically begin with structured data without predefined questions or tasks. In contrast, much of the existing research on large language models for data science (LLM4DS) continues to evaluate executional intelligence, treating a user-posed question as the primary objective for agentic models (Tang et al., 2025; Chen et al., 2025a; Zhou et al., 2025; Sun et al., 2025; Hong et al., 2025). Although recent work has moved towards more autonomous data analysis (Zhang et al., 2025c; Sundar et al., 2025), evaluation in these settings still relies on low level correctness checks and subjective assessments of report quality (Nascimento et al., 2024; Chen et al., 2025b; Zhang et al., 2025c). Even when explicit questions are removed, prompts typically include detailed instructions specifying what to investigate. Moreover, these settings are often small in scale, with agent-database interactions limited to fewer than a dozen steps. In parallel, deep research benchmarks evaluate agents that perform open-ended web search and report generation (Wong et al., 2025; Wan et al., 2025; Zhang et al., 2025d). While these benchmarks also aim to assess agency, they operate on unstructured web content and largely restrict tool use to issuing search queries and the evaluation remains challenging (Du et al., 2025).

Motivated by this gap, we formalise *Deep Data Research (DDR)*, an open-ended setting in which an agentic LLM is given only a structured database and a generic toolset, without predefined questions, objectives, or interaction limits. The agent must autonomously perform long-horizon exploration, formulate and test hypotheses, decide when to terminate exploration, and report the insights it discovers. Evaluation is conducted using a checklist derived from the free-form text components of the database, which enables factual verification of claims in the generated report and supports an interpretable, objective, and scalable assessment. Compared with prior work (Lu et al., 2025a; Zhang et al., 2025c), DDR enables fully open-ended data exploration while retaining rigorous and verifiable evaluation. We further introduce DDR-Bench, a benchmark that instantiates

[1]King's College London [2]Tencent [3]The Alan Turing Institute. Correspondence to: Wei Liu <wei.4.liu@kcl.ac.uk>, Yulan He <yulan.he@kcl.ac.uk>.

*Proceedings of the 43rd International Conference on Machine Learning*, Seoul, South Korea. PMLR 306, 2026. Copyright 2026 by the author(s).

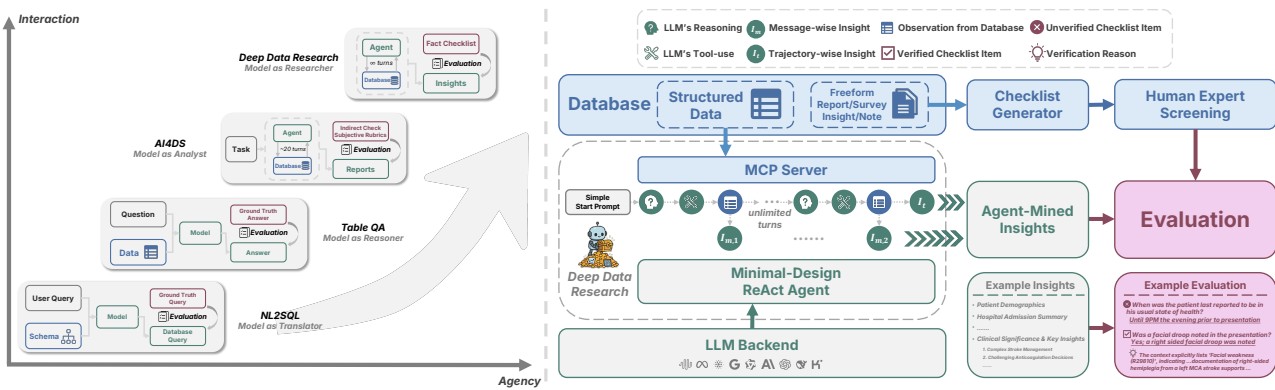

Figure 1. **Left**: Compared with previous tasks, *DDR* maximises exploration openness and agency, focusing on the direct evaluation of insight quality. **Right**: Overview of the DDR-Bench. Details of the pipeline are in Figure 2 and trajectory samples in Appendix M.

Deep Data Research over large real-world databases, as illustrated in Figure 1. Beyond reporting benchmark results, we conduct a systematic analysis of long-horizon interaction patterns, examining how agency steers models through Deep Data Research from multiple perspectives, as shown in Figure 4. In summary, this paper makes three contributions. **First, we formalise *Deep Data Research* (DDR)**, an open-ended agentic setting that isolates investigatory intelligence in structured data environments, requiring agents to autonomously explore data, generate and validate hypotheses, and determine when exploration should terminate, without predefined questions or objectives. **Second, we introduce DDR-Bench**, the first large-scale benchmark for DDR, which supports fully open-ended and long-horizon data exploration with objective and interpretable evaluation through checklist-based claim verification. **Third, we present a comprehensive evaluation and analysis of investigatory behaviour**, revealing persistent limitations in current frontier models. Our results show that progress in agentic LLMs requires not only longer error-free trajectories but also effective use of agency to identify information gaps, adaptively resolve uncertainty during long-horizon exploration, and maintain a globally stable exploration policy.

## 2. Method

### 2.1. Deep Data Research

As shown in Figure 1, the task of *Deep Data Research* can be formalised as $I = DDR(LLM, D, T)$, where, given a database $D$, an LLM iteratively queries the database using a tool set $T$ (e.g., SQL and Python), with no predefined limit on the number of interaction rounds. The process terminates only when the model judges that sufficient information has been gathered to produce a report containing insights $I$. The LLM only receives a simple start prompt specifying the task entity, which serves as the entry key for

database exploration, for example, "*Start analysing the user with userid=2048.*" In the first interaction round, the LLM is given basic metadata about the database. In each subsequent round, the model observes all previous results and then generates reasoning tokens $r$ and tool invocation tokens $t$. Then, these tool calls are executed on the database, and the resulting observations are returned to the model, denoted as $o$. Through this ReAct-style (Yao et al., 2023) interaction sequence $(r, t, o)$ over multiple rounds, agentic models have access to the full reasoning and action trajectory, with interleaved thinking and execution.

The model autonomously determines when to stop exploration and produces two types of insights $I = (I_m, I_t)$ that capture salient patterns and relationships discovered during the process. The first is message-wise insight $I_m$, in which the model is prompted (see Appendix L.2) to interpret the $(r, t, o)$ of each ReAct round as an insight paragraph. The second is trajectory-wise insight $I_t$, where the model self-terminates and reviews the whole history $(r_i, t_i, o_i)_{i=1}^{M}$ and synthesises it into a single report as the last step of the trajectory, where $M$ is the length of the trajectory.

### 2.2. Agent Design

We then construct DDR-BENCH, a benchmark and evaluation suite for deep data research. It adopts a simple agent scaffold that imposes minimal constraints on model behaviour. Its design is guided by three core principles. **1) No query or question is provided**. The benchmark is not structured as conventional input-output pairs; instead, it consists of a database paired with multiple prompt-checklist instances. Models are given only a minimal start prompt[1].

---

[1] A minimal start prompt refers to providing only the task entity, without giving any explicit goals. However, the system prompt still includes basic instructions such as role description and return format, as illustrated in Appendix L.1

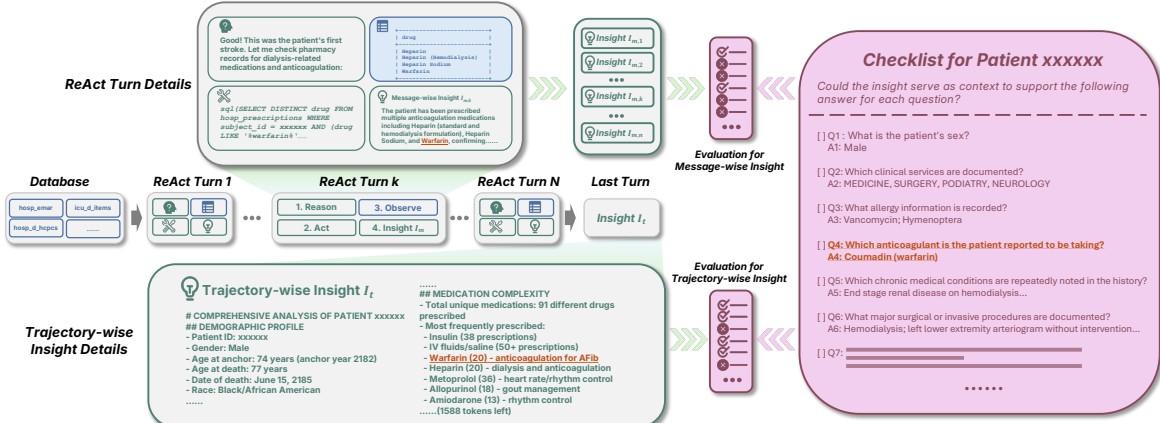

Figure 2. A case of Claude Sonnet 4.5's trajectory and evaluation checklist in the MIMIC scenario of DDR-Bench. Verified fact and supporting insights are underlined. See details of this trajectory in Figure A19. The patient id is anonymised.

**2) The agent framework is deliberately minimal**. The system prompt follows a lightweight *ReAct* style and excludes explicit workflow, memory, or planning modules. Only two fundamental data analysis tools are exposed through the standard Model Context Protocol (Anthropic, 2024), which are SQL and Python. Although complex agent frameworks have demonstrated strong performance on specific tasks, recent advances in agentic LLMs (Zhang et al., 2025b) suggest that many scaffolded capabilities can be internalised within the models themselves. This motivates DDR-Bench to focus on assessing model capabilities such as tool use or long-horizon reasoning, without confounding effects from external scaffolding. **3) Exploration is unrestricted**. No upper bound is imposed on the number of interaction rounds, and termination is determined autonomously by the model. Together, these design choices maximise model agency and position DDR-BENCH as a faithful benchmark for evaluating emerging *agentic LLMs*.

## 2.3. Data Collection and Construction

In constructing DDR-BENCH, we select three fundamentally distinct scenarios, with databases consisting of structured data and unstructured text. This integration enables the derivation of reliable checklists from the unstructured components, which are then used to assess the quality of insights obtained through autonomous exploration of the structured data. Scenarios in DDR-Bench include: 1) **MIMIC-IV** (Johnson et al., 2023), which is a large-scale de-identified EHR database. The *Hosp* and *ICU* modules are retained as structured tables, while the unstructured clinical narratives in the *note* module are used to derive checklists. A total of 100 patients are uniformly sampled with stratification by note count, ensuring coverage of both simple and highly complex clinical courses. 2) **GLOBEM** (Xu et al., 2022), which is a Sport and Exercise Psychology database that combines structured wearable sensor signals with partic-

Table 1. Key statistics of the DDR-Bench. A checklist item denotes a fact that is used to evaluate the quality of model-mined insights.

| Database | Records | Tables | Fields | Checklist Items |
|---|---|---|---|---|
| *MIMIC* | 200M+ | 29 | 318 | 774 |
| *GLOBEM* | 55K+ | 6 | 222 | 435 |
| 10-*K* | 3M+ | 5 | 5,832 | 849 |
| **Total** | **203M+** | **40** | **6,372** | **2,058** |

ipant surveys on social well-being and mental health. After sampling and filtering, 91 users with non-trivial temporal behaviour patterns are retained as task entities. Although GLOBEM does not include unstructured text, insight-level facts can still be derived from survey responses, such as comparisons of psychological and social well-being before and after the intervention. 3) **10-K**, which is constructed from annual reports of publicly listed United States companies obtained through the official publicly available SEC API [2]. The structured component comprises XBRL-formatted financial statements, while the unstructured component consists of cleaned textual sections that are closely aligned with the financial data. A total of 100 representative companies are selected, each with complex financial statements and more than 5, 800 distinct financial facts. Across the three domains, there are 291 task entities in total (shown in Table 1).

## 2.4. Evaluation

Evaluating deep data research is fundamentally challenging because models are expected to generate unconstrained insights through exploratory interaction rather than fixed answers to predefined questions. Existing evaluation paradigms for data exploration and report generation remain inadequate, as they embed implicit research objectives

---

[2]https://www.sec.gov/search-filings/edgar-application-programming-interfaces.

within detailed prompts that compromise the open setting and obscure the assessment of model agency, or depend on subjective report-level scores or indirect factual checks such as code execution validation, neither of which directly evaluates the faithfulness of reported insights to the underlying data. To overcome these limitations, we adopt a checklist evaluation framework over hybrid structured databases. For each database, verifiable factual statements are extracted from unstructured components using GPT-5 mini (OpenAI, 2025) and organised into a fact checklist. We apply GPT-5 mini and use this checklist to evaluate whether an insight offers sufficient contextual evidence to support each fact (See details in Appendix L.4). Human expert screening is performed to ensure that the mapping from the data domain to the fact domain is surjective, meaning that every checklist item can be supported by analysing some subset of the database. More than fifty domain experts participate in screening to confirm that each fact is reasonably explorable and inferable from the corresponding data. This process results in three data domains comprising 291 task entities and 2,058 verified checklist items, each linked to one or more queryable database components, ensuring that the checklist is objective, verifiable, and defined at the level of individual samples rather than through global subjective judgements. Checklist categories and representative examples from each domain are presented in Appendix §G and §H, and an evaluation example is shown in Figure 2. By relying on objective checklists rather than subjective rubrics, the evaluation directly targets verifiable data exploration and insight interpretation. We conduct extensive experiments to ensure the faithfulness (see Appendix §E) and trustworthiness (see Appendix §F).

## 3. Benchmark Results

We benchmark a broad set of proprietary LLMs (Anthropic, 2025; OpenAI, 2025; Comanici et al., 2025) and open-source LLMs (team, 2025; Team, 2025a;b; MiniMax AI, 2025; Yang et al., 2025a; Qwen et al., 2025; Grattafiori et al., 2024). We report both accuracy (§3.1) and novelty (§3.2). In addition to final results, we annotate sampled trajectories and summarise representative agent failure modes in Appendix D.

### 3.1. Checklist Accuracy

We report four types of accuracy, combining the average over all task entities (Sample-Averaged) or all checklist fact items (Item-Averaged) with message-wise insights $I_m$ and trajectory-wise insights $I_t$, as shown in Table 2. Multiple proprietary and open-source models were evaluated, and to date, only Claude 4.5 Sonnet achieves an average accuracy above 40%. While leading proprietary LLMs show broadly similar performance across most benchmarks, Claude stands

out on DDR-Bench, where its agentic capabilities enable it to surpass models such as GPT and Gemini. Top open-source models, including GLM, Kimi, and Minimax, approach the performance of proprietary LLMs. These findings indicate that deep data research tasks, which closely resemble real-world exploratory behaviour, remain far from saturated by current models.

### 3.2. Novelty Analysis

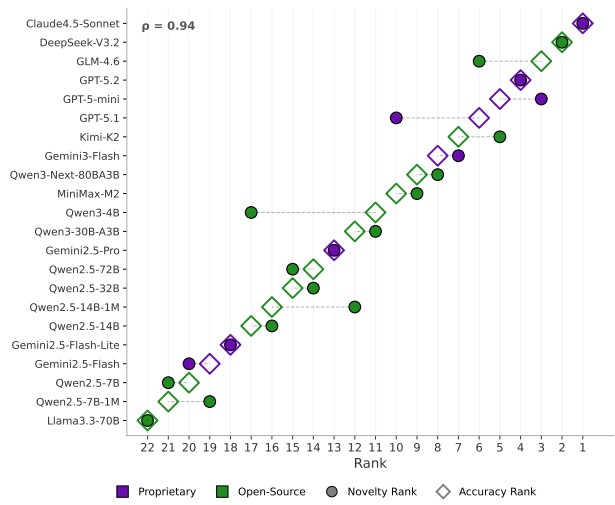

*Figure 3.* Ranking comparison between novelty and accuracy in the 10-K scenario, which shows high correlation.

Although DDR-Bench offers a more objective evaluation than prior open-ended data research benchmarks that rely heavily on subjective rubrics (Zhang et al., 2025c; Pérez et al., 2025; Zhang et al., 2025a), the open-ended nature of the task inevitably produces false positives, as no predefined checklist can exhaustively capture all valid insights. Manually annotating all model outputs is prohibitively expensive, operationally infeasible, and prone to subjective bias. Instead of eliminating such cases, we analyse false-positive novel insights using a pairwise comparison framework. For each model and task–entity pair, we extract message-wise insights $I_m$ that are not used in evaluating any checklist item and treat them as *novel insights*. We then use GPT-5-mini to conduct pairwise comparisons between two models on the same task and entity, judging which model produces more useful novel insights, with ties allowed (see Appendix §L.3). To mitigate position and identity bias, model identities are anonymised, and order is randomly swapped. Pairwise outcomes are aggregated into a global ranking using the Bradley–Terry model (Bradley & Terry, 1952).

Figure 3 illustrates the comparison between accuracy ranking and novelty ranking in the 10-K scenario, and the full ranking results are shown in Appendix A. The correlation coefficients for the MIMIC, 10-K, and GLOBEM scenar-

*Table 2.* Benchmarking results. The best results are highlighted in **bold**. Accuracy is defined as the proportion of checklist items verifiable from the model-mined insights, reported as either sample-averaged (over task entities) or item-averaged (over checklist items). We place no budget limit on the model's exploration, and report the distribution of exploration rounds for each model in §I.

| Models | Sample-Averaged Accuracy | | | | | | Item-Averaged Accuracy | | | | | | Overall Avg. |
|---|---|---|---|---|---|---|---|---|---|---|---|---|---|
| | Message-Wise Insights | | | Trajectory-Wise Insights | | | Message-Wise Insights | | | Trajectory-Wise Insights | | | |
| | MIMIC | GLOBEM | 10-K | MIMIC | GLOBEM | 10-K | MIMIC | GLOBEM | 10-K | MIMIC | GLOBEM | 10-K | |
| *Proprietary Models* | | | | | | | | | | | | | |
| Claude 4.5 Sonnet | **36.07** | **40.13** | **77.61** | 34.67 | **38.72** | 60.58 | 34.37 | 40.23 | **77.27** | 32.95 | **38.85** | 61.25 | **47.73** |
| GPT-5.2 | 28.85 | 38.81 | 44.89 | 32.49 | 38.15 | 41.09 | 27.26 | 38.39 | 44.99 | 30.49 | 38.39 | 41.22 | 37.09 |
| GPT-5.1 | 28.37 | 38.31 | 37.12 | **35.24** | 35.79 | 44.25 | 26.61 | 37.88 | 37.69 | **33.59** | 35.63 | 44.76 | 36.27 |
| GPT-5 mini | 30.02 | 35.86 | 46.82 | 27.86 | 31.54 | 37.12 | 28.81 | 36.09 | 46.35 | 26.36 | 31.72 | 36.77 | 34.61 |
| Gemini 3 Flash | 26.58 | 35.60 | 44.82 | 20.78 | 36.74 | 21.24 | 24.94 | 35.29 | 44.41 | 19.51 | 36.78 | 21.08 | 30.65 |
| Gemini 2.5 Pro | 21.51 | 33.77 | 24.48 | 20.00 | 35.62 | 15.57 | 19.51 | 33.79 | 25.68 | 18.48 | 35.40 | 16.14 | 25.00 |
| Gemini 2.5 Flash | 16.64 | 29.06 | 8.48 | 23.76 | 28.44 | 16.06 | 14.99 | 28.95 | 8.72 | 22.22 | 28.28 | 16.49 | 20.17 |
| Gemini 2.5 Flash-Lite | 17.19 | 26.63 | 19.45 | 17.96 | 24.03 | 9.01 | 16.10 | 26.90 | 19.32 | 17.18 | 24.14 | 9.42 | 18.94 |
| *Open-Source Models* | | | | | | | | | | | | | |
| DeepSeek-V3.2 | 28.98 | 38.46 | 60.08 | 30.57 | **38.46** | **38.15** | 27.00 | 38.16 | 60.66 | 28.29 | **38.62** | **38.16** | **38.80** |
| GLM-4.6 | 25.03 | **41.56** | 60.31 | 26.15 | 37.60 | 36.02 | 23.26 | **41.61** | 60.42 | 24.42 | 37.70 | 36.16 | 37.52 |
| Kimi K2 | **33.61** | 37.14 | 51.06 | **30.69** | 37.00 | 30.84 | **31.65** | 37.01 | 51.24 | **28.68** | 37.01 | 31.10 | 36.42 |
| MiniMax-M2 | 25.39 | 37.07 | 44.17 | 24.36 | 36.81 | 26.88 | 23.90 | 37.24 | 44.66 | 23.13 | 36.55 | 26.82 | 32.25 |
| Qwen3-Next-80B-A3B | 18.01 | 35.75 | 44.76 | 21.79 | 33.06 | 30.82 | 16.80 | 35.40 | 45.58 | 20.80 | 32.87 | 31.10 | 30.56 |
| Qwen3-30B-A3B | 21.67 | 35.73 | 42.44 | 18.75 | 37.25 | 14.38 | 20.03 | 35.63 | 42.33 | 18.22 | 37.01 | 14.13 | 28.13 |
| Qwen3-4B | 17.97 | 25.99 | 41.13 | 18.68 | 27.55 | 19.76 | 16.67 | 26.21 | 40.94 | 17.18 | 27.59 | 19.91 | 24.97 |
| Qwen2.5-72B | 15.65 | 28.83 | 27.13 | 16.82 | 25.38 | 13.42 | 14.34 | 28.74 | 27.56 | 15.50 | 25.52 | 14.02 | 21.08 |
| Qwen2.5-14B-1M | 16.75 | 28.75 | 22.69 | 13.68 | 24.29 | 10.13 | 15.50 | 28.80 | 23.56 | 12.66 | 24.14 | 9.78 | 19.23 |
| Qwen2.5-32B | 14.12 | 25.67 | 27.07 | 14.40 | 25.88 | 7.90 | 13.05 | 25.82 | 27.53 | 13.18 | 25.98 | 8.12 | 19.06 |
| Qwen2.5-14B | 15.93 | 25.56 | 18.91 | 14.51 | 26.47 | 10.60 | 14.86 | 25.59 | 19.18 | 13.44 | 26.67 | 10.12 | 18.49 |
| Qwen2.5-7B | 12.81 | 27.20 | 10.52 | 11.08 | 23.75 | 4.51 | 11.63 | 27.36 | 10.46 | 9.95 | 23.91 | 4.59 | 14.81 |
| Qwen2.5-7B-1M | 14.61 | 29.34 | 9.42 | 5.95 | 25.68 | 3.91 | 12.85 | 29.00 | 9.71 | 5.30 | 25.29 | 3.65 | 14.56 |
| Llama3.3-70B | 10.59 | 23.99 | 9.91 | 5.51 | 21.70 | 2.95 | 9.56 | 23.68 | 9.95 | 5.04 | 21.61 | 3.06 | 12.30 |

ios are 0.90, 0.94, and 0.72, respectively. Differences between the two rankings are small, especially among the top-performing models. Strong models consistently achieve high checklist accuracy and produce more useful novel insights, regardless of their quantity. This finding highlights a key property of the DDR-Bench evaluation framework. Although checklist-based assessment necessarily covers only a subset of all potential insights, it does not systematically undervalue models that attend to aspects beyond the checklist. Instead, DDR-Bench captures the dominant insight signal. Models with higher checklist scores also tend to generate novel insights that are judged to be more useful.

## 4. Investigatory Dynamics

We study **investigatory dynamics** in this section, which characterise how models behave under agentic test-time scaling in deep data research. This includes how sustained, multi-step interactions influence performance (§4.1), the structure of exploration patterns during investigations (§4.2), and when models decide to terminate an investigation (§4.3).

### 4.1. Scaling Analysis

We analysed model performance under test-time scaling using message-wise insights $I_m$. The final scaling curves represent the results averaged over all task entities, and the distribution of interaction turns is provided in Appendix §I. Test-time scaling is examined from three perspectives: *in-*

*teraction scaling*, *token scaling*, and *cost scaling*, as shown in Figure 4. Additionally, in Appendix B, we examine how training-time factors, including parameter scaling, long-context modelling, and mid-training, shape models' investigatory behaviour at test-time scaling.

**Interaction Scaling.** Model performance generally follows a sigmoid-shaped trajectory, which is expected. Furthermore, substantial variation appears in the timing and ceiling of improvement across models. Notably, higher-ceiling models often delay entry into the rapid improvement phase, avoiding early collection of practical information, as seen in Claude, GLM, and DeepSeek. Such patterns resemble a plan-then-act strategy, but examination of trajectories indicates that this behaviour does not arise from explicit planning, since DDR-Bench adopts a minimalism design with no planning prompts; also, the databases are typically too large for comprehensive upfront planning. Instead, it reflects *implicit planning* manifested through interaction dynamics: although no single response explicitly articulates a plan, the sequence of reasoning steps and function calls unfolds as if guided by a predefined plan.

**Token Scaling** Scaling by costed tokens, which include all LLM input and output tokens along the exploration trajectory, reveals a distinct pattern. Token usage per interaction varies across exploration stages, causing scaling curves to shift from a sigmoid shape to one that is initially flat and

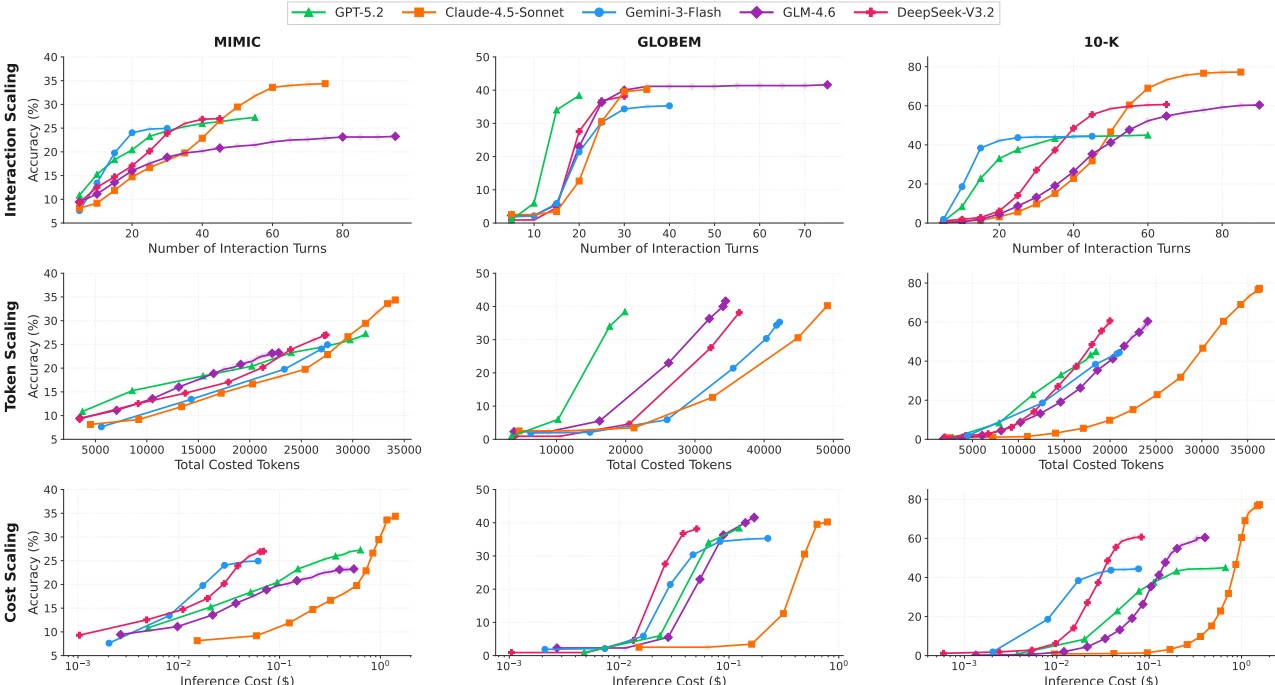

*Figure 4.* Inference-time scaling performance in DDR-Bench across different dimensions. The y-axis reports checklist accuracy. Beyond final accuracy, DDR-Bench provides rich test-time exploration information from different scaling dimensions, enabling detailed analysis of model agency behaviour. See details in §4.

then sharply increasing, with no clear saturation, underscoring the disproportionate value of late-stage tokens. Early interactions yield easily accessible insights through broad queries, whereas later gains rely on synthesising accumulated evidence and issuing a small number of highly targeted queries, marking a transition from breadth-oriented to depth-first exploration. Token consumption is largely dominated by environmental feedback, particularly database tool execution results, implying that the most valuable late-stage tokens are few and that performance improvements stem from deep, focused queries rather than exhaustive search. Case analyses further show that top-performing models concentrate on exploring and verifying specific hypotheses by issuing complex tool parameters while receiving minimal, often binary, feedback, as the verification logic is encoded in the query or code itself. This pattern is also reflected in tool call latency: the best-performing models exhibit moderate tool invocation time (see Appendix §K).

**Cost Scaling**   Model cost is determined by input and output token amounts, and serving expenses, enabling fair comparison of scaling curves in terms of the cost of intelligence. Claude is the most expensive model but consistently achieves the highest performance, while DeepSeek exhibits strong cost-effectiveness. The non-uniform horizontal axis reveals that intelligence gains often require exponential increases in cost. Optimising models through lower training

costs and more efficient inference-time scaling can shift the entire cost curve leftward by orders of magnitude. For instance, in the 10-K scenario, both model pairs follow similar scaling trends, yet at comparable performance levels, Gemini is an order of magnitude cheaper than GPT, and DeepSeek likewise outperforms GLM in cost efficiency. Efficient models also sustain a high rate of effective exploration while generating sufficient insights, captured by the valid insight ratio, defined as the proportion of interactions that yield genuine insights (see Appendix §J).

### 4.2. Exploration Patterns

To examine interaction behaviour in greater detail, we visualise model exploration patterns, as shown in Figure 5. The vertical axis reports *database coverage*, defined as the proportion of distinct fields accessed relative to the total number of available fields. The horizontal axis represents *exploration entropy*, which captures how uniformly a model distributes its access across fields during exploration. To make this quantity comparable across databases of different sizes, we adopt *Normalised Exploration Entropy*. Let $\mathcal{F} = f_1, f_2, \ldots, f_n$ denote the set of $n$ distinct fields accessed by a model, and let $c_i$ be the number of times field $f_i$ is accessed. The total number of accesses is $N = \sum_{i=1}^{n} c_i$, yielding an access probability $p_i = c_i/N$ for each field. The entropy of this access distribution is $H = -\sum_{i=1}^{n} p_i \log_2 p_i$ . We normalise this value by the

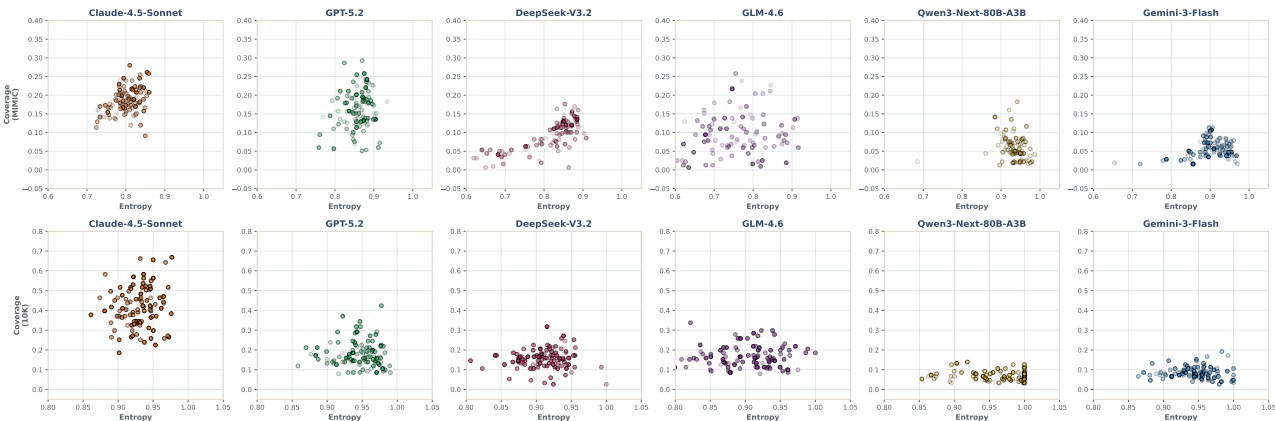

*Figure 5.* Exploration patterns of different models. The x-axis denotes exploration entropy, reflecting the depth of the model's search over the database, while the y-axis represents database coverage, indicating the breadth of the search.

maximum possible entropy $H_{\max} = \log_2 n$, which is attained under uniform access, obtaining a normalised exploration entropy in the range $(0, 1]$:

$$H_{\mathrm{norm}} = \frac{H}{H_{\max}} = \frac{-\sum_{i=1}^{n} p_i \log_2 p_i}{\log_2 n} \qquad (1)$$

This visualisation jointly captures exploration breadth and depth. Higher coverage reflects broader access to the database, while lower entropy indicates more concentrated, depth-oriented exploration. Each point corresponds to a single task entity, with transparency encoding checklist accuracy. GLOBEM is omitted because its focus on algorithm generation leads all models to access nearly all fields, obscuring meaningful field-level exploration patterns.

Despite differences in task settings and database scales, most models exhibit consistent exploration behaviour. Claude and GPT show particularly stable, balanced patterns with low variance across instances, suggesting robust internal exploration strategies, whereas GLM exhibits substantially higher variance, indicating heterogeneous trajectories. For weaker models, the visualisation reveals the sources of poor performance: Qwen and Gemini access a limited subset of fields, indicating insufficient breadth, while also showing high entropy, reflecting unfocused exploration.

### 4.3. Self-Termination

We further examine the model's self-termination behaviour, namely the point at which an LLM judges that sufficient information has been gathered and exploration should cease. All trajectories generated by the Qwen family are collected, and the probability of directly emitting exploration termination tokens after different numbers of turns is measured as $\frac{1}{N}\sum_{i=1}^{N} \log P(t_i \mid t_1, t_2, \ldots, t_{i-1}, T_{partial})$, where $N$ denotes the length of the finish tokens and $T_{partial}$ is a trajectory prefix containing varying numbers of turns. The results are shown in Figure 6. Clear differences emerge

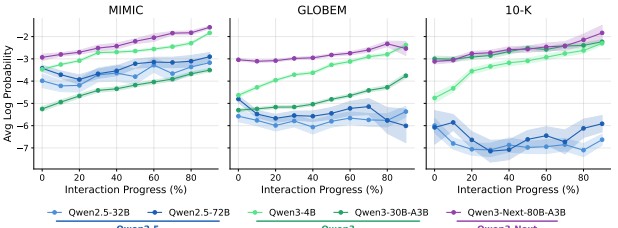

*Figure 6.* Self-termination visualisation on the Qwen family.

across model generations. Qwen3 and Qwen3-Next exhibit a consistently increasing probability, indicating growing confidence that a complete report can be produced as more information is accumulated, whereas the Qwen2.5 series shows pronounced fluctuations and remains uncertain about whether exploration can be terminated at the current step. Moreover, Qwen3-Next maintains higher confidence with lower variance throughout, suggesting that it has more confidence that exploration is progressing towards a more comprehensive and deeper report.

## 5. Module Analysis

We conduct an empirical analysis of several agent modules to examine how these capabilities interact with agentic LLMs to maximise effective agency in deep data research.

**Reasoning** We increase the reasoning budget of Qwen3-Next-80B-A3B, making it generate more reasoning tokens per interaction, and evaluate the effects. Trajectory analysis shows that explicit reasoning occurs mainly in initial rounds, with later steps dominated by tool calls, while some reasoning is implicitly encoded in call arguments, such as comments in generated code. This leads to very few explicit reasoning tokens, especially on 10-K and GLOBEM. As Table 3 shows, a higher reasoning budget increases reasoning

tokens and reduces interaction rounds, reflecting that richer reasoning per round enables more comprehensive queries. However, final performance fluctuates, revealing a trade-off between reasoning depth and interaction frequency, suggesting that reasoning and interaction should be dynamically adaptive rather than fixed.

*Table 3.* Comparison of reasoning-token usage, interaction efficiency, and performance under different reasoning budgets for Qwen3-Next-80B-A3B.

| Stat | Reasoning | 10-K | GLOBEM | MIMIC |
|---|---|---|---|---|
| Reasoning Tokens Per Turn | Default | 1.20 | 15.26 | 249.75 |
| | Longer | 357.78 ↑ | 397.11 ↑ | 417.65 ↑ |
| Interactive Turns | Default | 27.93 | 12.93 | 14.56 |
| | Longer | 11.89 ↓ | 9.41 ↓ | 11.49 ↓ |
| Accuracy - $I_m$ (%) | Default | 45.58 | 35.40 | 16.80 |
| | Longer | 36.40 ↓ | 36.78 ↑ | 16.67 ↓ |
| Accuracy - $I_t$ (%) | Default | 31.10 | 32.87 | 20.80 |
| | Longer | 37.34 ↑ | 33.10 ↑ | 20.67 ↓ |

**Memory** DDR-Bench uses a minimal ReAct Agent to evaluate a base model's intrinsic capabilities without external framework interference, providing the full agent trajectory without memory. To test memory effects, Qwen3-Next-80B-A3B was also evaluated with a standard agent memory mechanism that summarises past trajectories into a local, readable and writable file. As Table 4 shows, this setup produces unpredictable behaviour: while the memory reduces long-context interference, it often triggers more aggressive tool use, larger per-round data reads, and earlier exploration termination. Overall, memory does not reliably improve accuracy, and agent frameworks require careful, case-specific design to provide gains.

*Table 4.* Effect of a long–short-term memory mechanism on the behaviour and performance for Qwen3-Next-80B-A3B.

| Scenario | Memory | Statistics | | | Accuracy (%) | |
|---|---|---|---|---|---|---|
| | | Traj Tok. | Turns | Memory Tok. | Traj | Msg |
| 10-K | – | 21,256.20 | 27.93 | — | 31.10 | 45.58 |
| | ✓ | 24,328.14 ↑ | 18.30 ↓ | 2,872.28 | 25.21 ↓ | 37.34 ↓ |
| GLOBEM | – | 23,140.60 | 12.93 | — | 32.87 | 35.40 |
| | ✓ | 26,184.59 ↑ | 20.52 ↑ | 2,008.73 | 35.86 ↑ | 35.86 ↑ |
| MIMIC | – | 19,574.87 | 14.56 | — | 20.80 | 16.80 |
| | ✓ | 23,650.02 ↑ | 18.28 ↑ | 1,670.57 | 14.34 ↓ | 15.63 ↓ |

**Reactive vs. Proactive** DDR-Bench uses a query-free, proactive exploration paradigm to evaluate LLM agency. We also implement a reactive variant, converting each checklist item into explicit user queries for goal-directed exploration. As Table 5 shows, Qwen3-Next-80B-A3B achieves higher accuracy under this variant, indicating that clear objectives significantly aid performance. This contrast underscores that autonomous goal identification is a far more challenging test of agency. While explicit goals show the tasks are solvable, improvements remain partial, and performance can degrade

in some cases, reflecting inherent model limitations even when objectives are well defined.

*Table 5.* Comparison of proactive (query-free) and reactive (goal-directed) exploration on Qwen3-Next-80B-A3B.

| Modes | MIMIC | 10-K | GLOBEM | Avg |
|---|---|---|---|---|
| *Proactive(DDR-Bench)* | | | | |
| Message-wise | 16.80 | 45.58 | 35.40 | 32.59 |
| Trajectory-wise | 20.80 | 31.10 | 32.87 | 28.26 |
| *Reactive* | | | | |
| Overall | 27.13 ↑ | 70.55 ↑ | 31.95 ↓ | 43.21 ↑ |

# 6. Related Work

**LLM and Agent for Data Intelligence** LLMs are increasingly positioned as data analysts and data science agents, reshaping how humans query and manipulate data (Tang et al., 2025; Chen et al., 2025a; Zhou et al., 2025; Sun et al., 2025; Qiao et al., 2025; Qiu et al., 2026). Early work mainly treats data intelligence as answering user specified queries, through table question answering and text to SQL over benchmarks such as WikiTableQuestions, FeTaQA, HybridQA, Spider, and BIRD (Pasupat & Liang, 2015; Nan et al., 2022; Chen et al., 2020; Yu et al., 2018; Li et al., 2023). Subsequent systems move from answers to executable analysis code and visualisations, as in DS 1000 and LLM4DS (Lai et al., 2023; Nascimento et al., 2024), and introduce agents specialised for plotting and database interaction (Yang et al., 2024; Xue et al., 2023; Wang et al., 2020; Pan et al., 2025). More recent work begins to automate broader data science workflows and proposes agent based benchmarks such as LAMBDA, DataSciBench, and DABStep to evaluate end-to-end behaviour (Li et al., 2025; Hong et al., 2025; Zhang et al., 2025a; Salemi et al., 2025; Egg et al., 2025; Guo et al., 2024a; Hollmann et al., 2023; Testini et al., 2025). Systems like DeepAnalyze and I2I STRADA go further by letting an agentic LLM conduct open-ended workflows over databases and produce narrative reports (Zhang et al., 2025c; Sundar et al., 2025). Across this line of work, however, either the data problem is still typically posed in advance and evaluation emphasises subjective metrics of report quality, or the report evaluation is transformed into closed-form questions by prompting LLMs. In contrast, DDR-Bench starts from the data alone and asks what an agent can discover, measuring its ability to mine concrete, fact-checked insights rather than simply data statistics.

**Proactive Agentic AI** Agentic AI studies language model agents that reason, plan, and act through tools and multi-step control, with frameworks such as ReAct and survey work systematising architectures and evaluation (Yao et al., 2023; Plaat et al., 2025; Wang et al., 2025a; 2024; Guo et al., 2024b; Yehudai et al., 2025). Within this space, a

significant theme is *proactivity*. User-facing agents in visual analytics, mobile interaction, video viewing, and professional workflows monitor activity streams and decide when to intervene to anticipate needs while avoiding unnecessary disruption (Zhao et al., 2025; Wen et al., 2025; Yang et al., 2025b; Wang et al., 2025c; Lu et al., 2025b; Zhang et al., 2024). In these settings, the environment is a user-centred trace. Proactivity is defined as the inference and fulfilment of the user's immediate goals. From a more abstract perspective, however, a user activity stream is just one kind of data, and "user intent" is just one kind of latent insight that can be inferred from it. A complementary line of work pushes toward open-ended investigation: curiosity inspired and intrinsically motivated methods encourage agents to seek novel states or patterns, and deep research benchmarks ask agents to conduct unconstrained research over the web using search and browsing tools (Dai et al., 2025; Wong et al., 2025; Wan et al., 2025; Yao et al., 2026; Zheng et al., 2025). These efforts implicitly adopt a more general notion of proactivity as autonomous insight seeking. Still, their evaluations usually collapse behaviour into global scores for answers or long reports, often using language models as judges. DDR-Bench builds on this generalised view. It treats any data environment, rather than the user alone, as the primary object of investigation, views user intent inference as just one special case of insight discovery, and instantiates this perspective in a setting where agents must decide for themselves what to investigate and are scored by a checklist-based, per-sample fact-checking of the insights they claim to have found.

**Open-Ended Data Analysis** Prior work has explored the use of LLMs for open-ended data analysis, yet a principled methodology for benchmark construction remains largely absent. Vykhopen et al. (2025) relies on indirect evaluation signals, such as report writing time and qualitative comparisons against a human expert's report, but it does not introduce automated and direct metrics for assessing the quality of extracted insights. Lei et al. (2025) adopts subjective evaluation by LLMs together with multi-level rubrics, and additionally collects baseline reports to compute GSB. However, it still does not directly evaluate insight quality. More importantly, it specifies detailed data exploration objectives for each test case, which means the setting is not truly open-ended data analysis. Sahu et al. (2024) alters the data distribution by manually injecting anomalous patterns as predefined insights to be discovered, resulting in a setup that more closely resembles information retrieval than genuine open-ended analysis. Egg et al. (2025); Zhang et al. (2025c) employ LLM-as-a-Judge and relies on subjective or indirect checklists, such as content relevance, professionalism of organisation and formatting, and the presence of data visualisations. Islam et al. (2024) similarly uses LLM-as-a-Judge to conduct pairwise comparisons along dimensions such as Informativeness, Clarity and Coherence. Gupta et al. (2025) automatically generates questions using LLMs, and then applies LLMs as judges for scoring, which again falls short of a fully open-ended setting. Overall, existing benchmarks exhibit varying degrees of limitation in terms of openness, data scale, and whether insight quality is evaluated directly. Their analyses also concentrate primarily on final accuracy, and therefore cannot provide fine-grained analyses of exploration trajectories in the way that DDR-Bench does.

# 7. Conclusion

DDR-Bench establishes a rigorous framework for evaluating investigatory intelligence, revealing that the transition from reactive execution to proactive discovery relies less on external scaffolding or simple parameter scaling, and more on intrinsic exploration strategies that balance breadth with targeted reasoning. Our findings highlight a critical distinction between executional and investigatory capabilities, observing that effective agency emerges from implicit planning and the autonomous determination of termination criteria rather than rigid workflows. As the field advances, this underscores the necessity of moving beyond query-response alignment toward training paradigms that cultivate end-to-end autonomy, enabling agentic LLMs to not only answer but proactively define goals and pursue them to derive useful insights from large-scale data.

# Acknowledgments

This work was supported in part by the UK Engineering and Physical Sciences Research Council through a Turing AI Fellowship (grant no. EP/V020579/1, EP/V020579/2) and the Prosperity Partnership scheme (grant no. UKRI566). Wei is supported by a PhD studentship provided by King's College London (KCL). The authors acknowledge the use of Computational Research, Engineering and Technology Environment (CREATE) at KCL, and Inkfish through the EMBRACE research programme.

# Impact Statement

This paper introduces DDR-Bench to advance the field of autonomous data science and investigatory intelligence.

**Data Compliance and Privacy.** Our benchmark integrates diverse real-world datasets. We strictly adhere to all relevant data use agreements and ethical regulations, particularly for the medical subset, which utilises fully de-identified data accessed under credentialed requirements. No personally identifiable information is processed, ensuring compliance with standard privacy protocols. To ensure openness and transparency, we provide the complete code for running the agents and performing evaluation in the supplementary materials, and we will release this code publicly in the future. Sensitive datasets such as MIMIC and GLOBEM will be made available securely via their hosting platform, PhysioNet, in the form of derived datasets, with access granted only to researchers who have completed the required security certification.

**Reliability and Traceability.** We conducted a comprehensive analysis of model hallucinations, finding them to be low in our experimental settings (see Appendix E). To further mitigate risks associated with autonomous analysis, DDR-Bench is designed with **full traceability** as a core principle. Every generated insight is explicitly linked to a specific analysis turn, mapping the claim directly to the executable query (SQL/Python) and the raw environment observation. This structure ensures that all agent-derived conclusions are verifiable and grounded in factual evidence.

**Societal Implications.** While this work aims to democratize data science, we acknowledge the risk of automation bias. These agents are designed to assist human experts by handling laborious exploration, not to replace human judgment in high-stakes decision making.

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

# A. Novelty Analysis

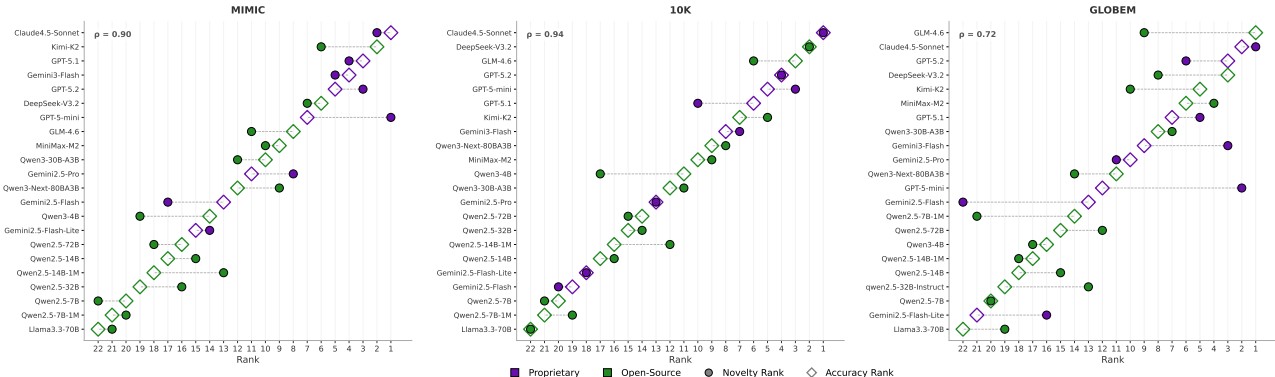

*Figure A1.* Ranking correlation between novelty and accuracy on Proprietary and Open-Source LLMs. Circles denote the novelty rank, and diamonds denote the accuracy rank. Models are ordered by accuracy rank in the figure. All three scenarios present high correlation.

Figure A1 shows the novelty ranking and accuracy ranking comparison among all three scenarios. All scenarios present a high correlation between novelty ranking and accuracy ranking.

# B. Study on Training Factors

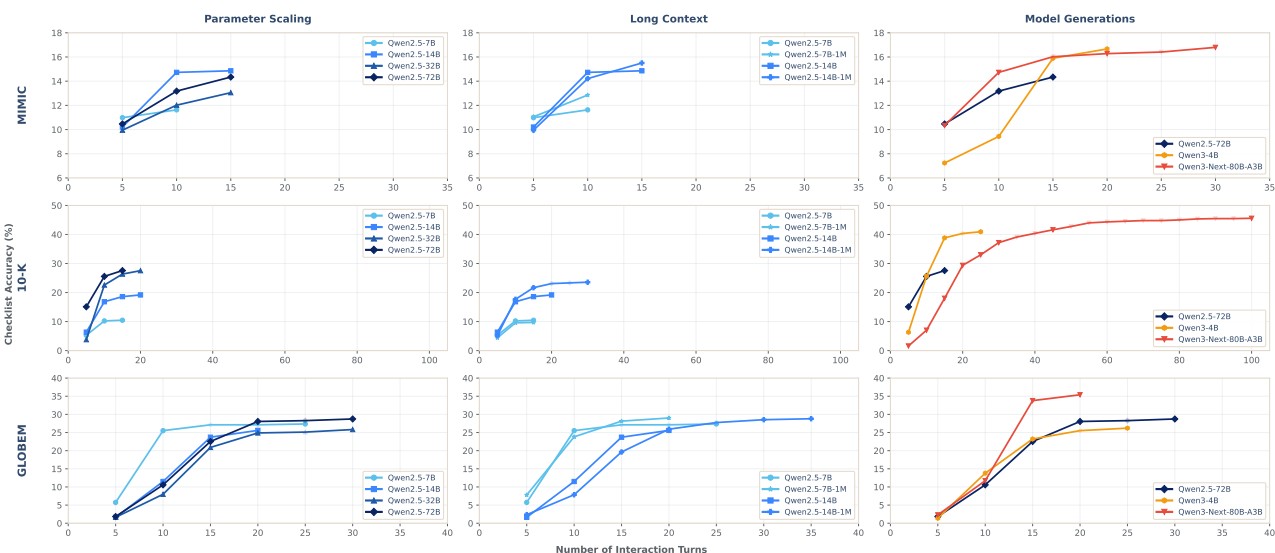

*Figure A2.* Training-time factors study within the Qwen family. From left to right, the three columns examine inference-time scaling performance across all scenarios for models with different parameter scales, different context optimisation methods, and different model generations with different training strategies.

We further analyse how different training-time technical choices would affect the model's investigatory intelligence in the Qwen family (Qwen et al., 2025; Yang et al., 2025a). Qwen offers a wide range of open-source models across multiple versions and scales, which makes it suitable for analysing within a single model family. The results are shown in Figure A2.

We first examine the effect of parameter scaling. Increasing model size yields only marginal improvements in accuracy. Even a tenfold increase in the number of parameters results in less than a 3% gain in final accuracy, and in several settings, smaller models even outperform their larger counterparts. In the case of Qwen2.5 models without explicit agent-oriented training, scaling parameters alone do not enhance proactive exploration and therefore fail to meaningfully strengthen agency.

We then compare the long-context variants of Qwen2.5 at the 7B and 14B scales. Extending the context window does not consistently improve overall performance, nor does it lead to a systematic increase in the number of interaction rounds.

Combined with the parameter scaling results above, this suggests that although larger model sizes and longer context lengths are often assumed to facilitate multi-turn agent behaviour, neither factor plays a decisive role in determining agentic capability in deep data research.

Finally, we examine models from different generations, namely Qwen3 and Qwen3-Next, with activated parameter sizes of 4B and 3B, respectively. Despite having fewer activated parameters, both models exhibit a clear increase in the number of exploration rounds on both MIMIC and 10-K. On GLOBEM, the number of exploration rounds remains largely unchanged; however, the models achieve a higher performance ceiling. Taken together, these cross-generation results indicate that improvements in agentic behaviour are primarily driven not by scale, but by a systematic emphasis on reasoning and agentic ability throughout the training pipeline, encompassing both pre-training and post-training stages (Yang et al., 2025a).

## C. Agent Framework Ablations

We intentionally simplify the agent to evaluate model capabilities in a clean testbed; optimising agent scaffolding for high scores would confound assessment. DDR-Bench supports flexible tool extensions via the Model Context Protocol (MCP) for future agent benchmarking.

We presents agent ablations, where memory modules often destabilise performance. Table A1 provides additional experiments on Qwen3-30B-A3B, Qwen3-4B, and GPT-5-mini across three framework configurations: Planning (Plan-and-Execute (Wang et al., 2023)), Memory (CoALA (Sumers et al., 2023)), and Multi-agent (AutoGen (Wu et al., 2024)). Results reveal that complex agents mostly degrade performance relative to the ReAct baseline, except for minor planning benefits in certain settings. Analysis suggests that complex frameworks affect model confidence, leading to premature self-termination or over-/under-thinking.

*Table A1*. Performance comparison of agent framework configurations across datasets and models. Scores are reported for ReAct (baseline), Plan-and-Execute (+Plan (Wang et al., 2023)), CoALA (+Memory (Sumers et al., 2023)), and AutoGen (+Multi-Agent (Wu et al., 2024)). Higher is better.

| Dataset | Model | ReAct | +Plan | +Memory | +Multi-Agent |
|---------|-------|-------|-------|---------|--------------|
| MIMIC | Qwen3-4B | 16.67 | 8.14 | 11.46 | 4.44 |
| | Qwen3-30B-A3B | 20.03 | 12.27 | 13.57 | 9.04 |
| | GPT-5-mini | 28.81 | 23.67 | 22.22 | 12.66 |
| 10-K | Qwen3-4B | 40.94 | 14.25 | 17.43 | 26.50 |
| | Qwen3-30B-A3B | 42.33 | 47.59 | 37.10 | 31.80 |
| | GPT-5-mini | 46.35 | 49.82 | 45.35 | 30.04 |
| GLOBEM | Qwen3-4B | 26.21 | 22.76 | 23.45 | 22.30 |
| | Qwen3-30B-A3B | 35.63 | 25.75 | 22.30 | 23.91 |
| | GPT-5-mini | 36.09 | 28.05 | 28.72 | 25.64 |

## D. Failure Modes

We sampled and manually annotated the failure modes of models. We collect evaluation results from all models across all scenarios, then extract them at the granularity of checklist items. From the items evaluated as incorrectly supported by the model's insight, we randomly sampled instances and linked each item to its corresponding trajectory, $I_t$, and $I_m$. We then manually annotated the reasons why the insights didn't correctly support the fact item. In total, 206 items were annotated.

We categorise the main errors into eight primary classes, which are discussed below. **1) Failure in Exploration**, such as errors arising from *insufficient breadth* or *insufficient depth* during database exploration. Limited breadth means that the model did not cover the necessary data sources, such as relevant tables or key data fields. It is mainly caused by premature self-termination of the exploration. Limited depth, by contrast, occurs when the model finds relevant data but does not delve deeper and develop more complex queries to extract more complex insights. **2) Poor Data-to-Insight Ability**, which reflects the deficiencies in the pipeline from raw data to statistics and ultimately to insights. The LLMs may stop at descriptive statistics without further interpretation (*superficial analysis*), draw wrong conclusions from the statistics (*insight misinterpretation*), or, on the contrary, engage in *over-reasoning*, imposing strong assumptions or causal claims that are not

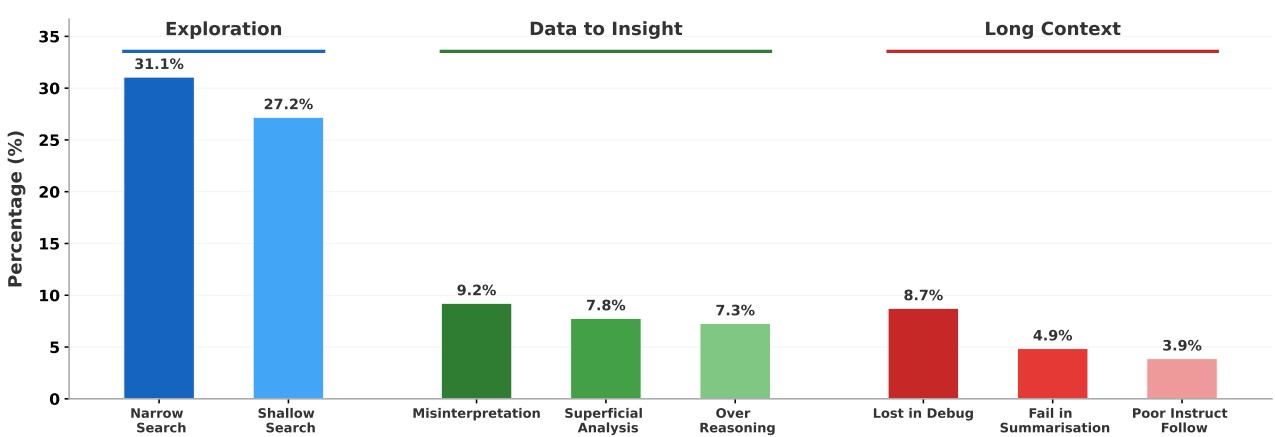

*Figure A3.* Distribution of manually annotated error types across models and task scenarios.

supported by the data. **3) Lost in the Context** is particularly prevalent among smaller LLMs, that trajectories are polluted by repetitive or unproductive behaviours, such as repeated *debugging in function calls*, failures in *instruction following*, or making summaries in $I_t$ but missing key points discovered during exploration (*fail in summarisation*).

As shown in Figure A3, our findings revealed that 58% of errors stemmed from insufficient exploration, both in terms of breadth and depth. This imbalance in exploration often leads to suboptimal results, regardless of the model's overall capability. Additionally, around 40% of the errors were attributed to other factors. For more powerful models, over-reasoning was common, where the model made assumptions not fully supported by the data. In other cases, models misinterpreted the insights, such as mistaking a downward trend for an upward one. Less capable models, on the other hand, tended to make more fundamental errors, such as repeatedly debugging or struggling with missing data, which could disrupt the overall coherence of the analysis.

## E. Hallucination Evaluation

*Table A2.* Hallucination rates (%) across models in DDR-Bench, measured as the proportion of insights containing factual but unfaithful information that are not derivable from the provided inputs.

| Model | 10-K | GLOBEM | MIMIC | Average |
|---|---|---|---|---|
| Claude 4.5 Sonnet | 4.98 | 1.11 | 6.15 | 4.08 |
| Gemini 2.5 Pro | 2.56 | 0.0 | 0.0 | 0.85 |
| Gemini 3 Flash | 1.82 | 0.0 | 0.0 | 0.61 |
| GLM-4.6 | 3.33 | 1.61 | 0.0 | 1.65 |
| GPT-5.2 | 0.0 | 0.0 | 2.86 | 0.95 |
| GPT-5 Mini | 0.93 | 0.0 | 2.38 | 1.10 |
| Kimi-K2 | 5.56 | 5.71 | 0.0 | 3.76 |
| MiniMax-M2 | 7.69 | 0.0 | 9.09 | 5.59 |
| Qwen3-Next 80A3B | 7.95 | 3.85 | 5.26 | 5.69 |
| DeepSeek V3.2 | 5.30 | 0.0 | 0.0 | 1.77 |

We conducted sampling at the granularity of insights for labelling the possible hallucination, namely, the model generates a factually correct insight with limited or wrong observations from the database. We select from the most recently released models within each model family, as these models are most likely to contain data contamination that could lead to hallucinations. In total, 1,850 insights were randomly sampled for annotation. We pair these insights with all inputs used to produce them, namely the system prompt, model reasoning, function calls, and the database results returned after execution, and checked whether any insight contained facts that were correct but could not be inferred from the available inputs. If a model had memorised relevant information due to data contamination, such cases would manifest as unfaithful but factual hallucinations, thereby distorting benchmarking results. We computed the proportion of insights exhibiting this behaviour, as reported in Table A2.

For most models, the hallucination rate is zero, while for the remaining models it stays below 5%. Many of these cases involve highly fine-grained entity-level hallucinations that do not affect overall insight judgment or benchmarking outcomes. A common pattern in the 10-K setting is the attribution of financial changes to real-world background events such as acquisitions, even when such information is not supported by the input data. This does not affect evaluation because scoring is based solely on conclusions derivable from the data, and external real-world associations receive no additional credit. Another frequent pattern appears in the MIMIC setting, where medication lists are incomplete due to the use of `LIMIT` in SQL outputs, and models infer additional drugs based on the diagnosed condition and common treatment combinations. While this behaviour likewise does not change evaluation scores, it poses potential risks in healthcare contexts. To mitigate such risks, each insight in DDR-Bench is explicitly linked to its full ReAct trace, including reasoning, tool calls, and execution results. This enables practitioners to verify references for each insight in real-world deployments.

We observe that hallucinations are largely absent in the GLOBEM dataset, likely because its structured and unstructured components originate from distinct domains, namely wearable data and psychological surveys, making cross-domain memorisation less applicable. Overall, hallucination rates on DDR-Bench are very low and have minimal impact on benchmarking results. Although larger models may theoretically retain more memorised knowledge, they also exhibit stronger instruction following and adhere more reliably to system prompts that restrict analysis to patterns observable in the provided data.

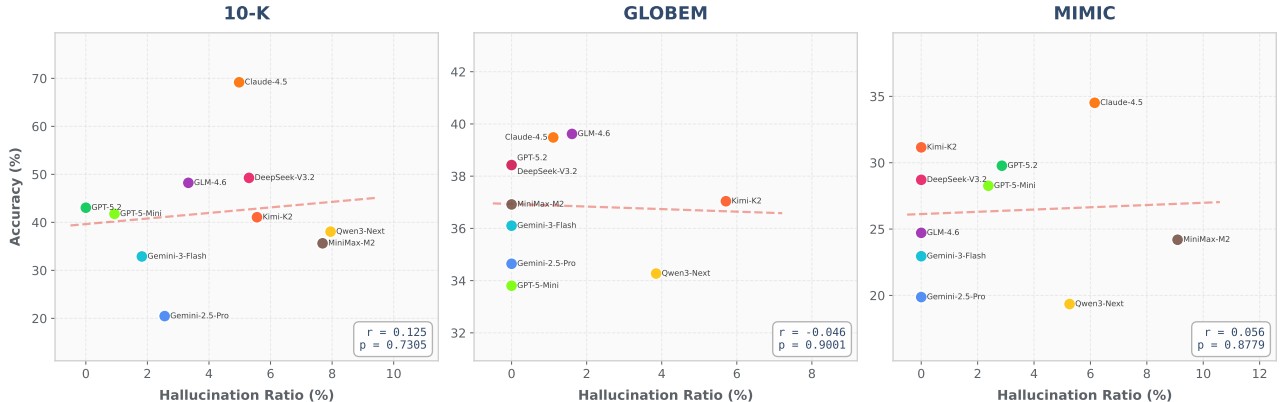

*Figure A4.* Hallucination-Accuracy Correlation.

Figure A4 shows that in the 10K and MIMIC settings, hallucination rates show extremely weak and non-significant positive correlations with accuracy, with correlation coefficients of 0.125 and 0.056, and $p$-values of 0.8779 and 0.7305, respectively. In the GLOBEM setting, the hallucination rate exhibits an extremely weak and non-significant negative correlation with accuracy, with a correlation coefficient of -0.046 and a $p$-value of 0.9001. Overall, the hallucination rate is essentially uncorrelated with final accuracy. It indicates that there is a very low risk that data contamination causes models to disregard actual database interaction results and inflate performance metrics.

## F. Trustworthiness on the LLM-as-a-Checker

*Table A3.* Stability and reliability of LLM-as-a-Checker evaluation in DDR-Bench across repeated runs and scenarios.

| database | Insight | Stability | | | | | Reliability |
|---|---|---|---|---|---|---|---|
| | | Max | Min | Mean | $\sigma$ | CV | Macro F1 |
| 10-K | $I_m$ | 34.05 | 31.18 | 33.05 | 1.12 | 3.38% | 92.03% |
| | $I_t$ | 15.05 | 13.62 | 14.27 | 0.64 | 4.49% | 90.05% |
| GLOBEM | $I_m$ | 28.79 | 27.27 | 27.88 | 0.83 | 2.98% | – |
| | $I_t$ | 28.03 | 26.52 | 27.42 | 0.63 | 2.31% | – |
| MIMIC | $I_m$ | 17.88 | 16.97 | 17.39 | 0.46 | 2.64% | 89.88% |
| | $I_t$ | 16.47 | 15.00 | 15.47 | 0.57 | 3.71% | 93.97% |

Because DDR-Bench adopts an LLM-as-a-Checker evaluation paradigm, we follow a methodology closely aligned with HealthBench (Arora et al., 2025) and conduct a systematic assessment of its stability and reliability. We emphasise that, in

DDR-Bench, the LLM evaluates model-generated insights by matching them against objective ground truth. It does not operate as an LLM-as-a-Judge in settings without ground truth, nor does it rely on subjective, rubric-based criteria. This design choice is intentional and ensures a high degree of objectivity and fairness in the evaluation process.

Specifically, we randomly sample about 10% of the data across all models and scenarios and repeat the evaluation five times. As reported in Table A3, the coefficients of variation across all settings are consistently below 5% for both insight types, demonstrating strong score stability under repeated evaluation. To further assess reliability, we manually annotate sampled trajectories from the MIMIC and 10-K scenarios and compare human annotations with LLM-generated scores. We do not conduct this comparison for GLOBEM, as it involves a closed-form evaluation and does not require an additional reliability check. The resulting macro F1 scores are consistently around 90%. These results provide strong empirical evidence that the LLM-as-a-Checker employed in DDR-Bench is both stable and reliable.

We further analyse the remaining discrepancies between LLM-as-a-Checker and human judgment. Two primary sources of inconsistency are identified. First, the LLM tends to apply overly strict criteria, requiring insights to explicitly include specific numerical values or statistics. Although these numbers are correctly retrieved by the models, they are often omitted in the final insights because the models are instructed to describe the underlying phenomena rather than enumerate individual figures. Second, in a small number of cases, the LLM produces correct intermediate reasoning but arrives at an incorrect final judgement.

## G. Checklist Statistics

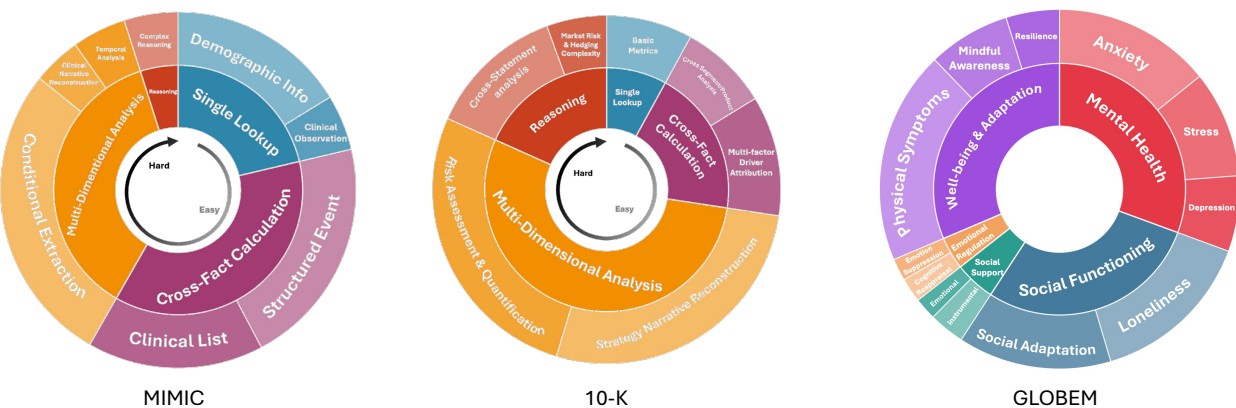

*Figure A5.* Distribution of checklist items.

Figure A5 shows the distribution of checklist items across the three DDR-Bench scenarios. For MIMIC and 10-K, the checklists are free-form. The outer ring groups items by question type, while the inner ring categorises the data analysis capabilities required of LLMs. Simple items, such as demographic information extraction in MIMIC or basic metric extraction in 10-K, require only a single lookup that can be completed with a straightforward SQL SELECT, without complex instructions or computation. In contrast, the most demanding items, such as analysing a patient's pathology by integrating multiple surgeries, diagnoses, and medications across departments, or assessing market risk by combining ten year financial trends with recent capital flows, require multi-table reasoning at the highest level of complexity. For GLOBEM, the checklist items are closed-formed. Numerical responses are extracted from each user's surveys, and questions are constructed by comparing changes in the same items before and after the wearable experiment, for example, whether the user's depression status improved after the experiment. The answers are closed-formed, with three options indicating improvement, no improvement, or no substantial change. When sufficient insights are derived from the wearable data, the LLM checker can answer these trend questions correctly based on the inferred evidence. This temporal comparison avoids pointwise prediction of absolute survey scores for social well-being or psychological status, which are highly subjective and exhibit large variance, and also avoids framing the task as numerical regression, which is not the aim of insight report generation. For GLOBEM, the outer and inner rings correspond to major and minor categories of survey questions. Required data modelling capabilities are not shown in the inner ring, as there is no ground truth modelling approach, and LLMs may flexibly analyse wearable data using Python code and any suitable algorithms.

# H. Checklist Example

Figure A6, A7 and A8 each present a checklist sample from one of the three scenarios in DDR-Bench, namely MIMIC, 10-K, and GLOBEM. MIMIC and 10-K involve open-ended question answering, whereas GLOBEM adopts a multiple-choice format with three fixed options: better, worse, and remained the same. Notably, for open-ended questions, we do not ask a judge LLM to first answer the question based on the agent's collected insights and then compare the answer with the ground truth. Instead, as illustrated in Figure 2, we provide the checker LLM with the insights, the question, and the ground truth, and ask it to assess whether the insights can support the given ground truth. This design serves two purposes. First, it avoids the difficulty of normalising open-ended answers for direct comparison. Second, it shifts the focus to whether the collected insights substantively support the underlying facts, rather than relying on exact matching of surface descriptions.

---

**Checklist Example in the MIMIC Scenario**

Q1: What is the patient's sex?
A1: Female

Q2: Which clinical services are documented?
A2: OBSTETRICS/GYNECOLOGY and MEDICINE

Q3: What allergies are listed?
A3: Shrimp; Ondansetron

Q4: Was a laparoscopic left salpingo-oophorectomy performed?
A4: Yes. A laparoscopic left salpingo-oophorectomy (LSO) is documented.

Q5: Was a single-balloon enteroscopy performed or planned?
A5: Yes. Single-balloon push enteroscopy is documented as a major invasive procedure.

Q6: Was pelvic washings performed during surgery?
A6: Yes. Pelvic washings are documented (conversion to ex-lap with LSO and pelvic washings).

Q7: Did the patient undergo an ovarian detorsion?
A7: Yes. Laparoscopic ovarian detorsion (left detorsion) is documented.

Q8: What did the final pathology of the ovarian tissue show?
A8: Adenocarcinoma involving ovarian tissue.

Q9: What diagnosis is listed as the primary oncologic diagnosis?
A9: Metastatic gastric adenocarcinoma

Q10: What histologic subtype of gastric cancer is mentioned?
A10: Signet ring gastric adenocarcinoma

---

*Figure A6.* Checklist Example in the MIMIC Scenario.

# I. Interaction Turn Distribution

Figure A9, A10 and A11 show the distributions of interaction rounds across models and scenarios. Almost all distributions are approximately normal, except for a small number of runs in which exploration failed to terminate, and the models repeatedly entered debugging loops. This behaviour was observed for a few models, such as Qwen3-30B-A3B and GLM4.6, and these runs were forcibly stopped at 100 rounds. We ignore these abnormal data points when drawing figures.

Stronger models tend to explore for more rounds without external prompting. Knowledge-intensive databases such as 10-K

Checklist Example in the 10-K Scenario

Q1: How has regulatory risk affected AEP's ability to recover its costs through rates?
A1: Regulatory risk has remained elevated recently — if regulators do not authorize timely rate adjustments, AEP could face reduced cost recovery and materially lower earnings (significant downside risk to results).

Q2: How does regulation affect AEP's rates and recovery of costs?
A2: Regulation remains central: rates are largely cost-of-service or formula-based and include mechanisms (fuel/purchased power recovery, trackers, formula rates) that enable more timely cost recovery; both state commissions and the FERC play active oversight roles, and recent management focus has been on alternatives to reduce regulatory lag.

Q3: What were the main drivers of the company's improved earnings in 2024?
A3: Earnings increased materially in 2024 driven mainly by favorable IRS PLRs that changed retail NOLC ratemaking, supportive rate proceedings, higher transmission-related revenues and increased sales volumes; these gains were partly offset by higher operating costs and certain unfavorable prior regulatory rulings. The improvement was concentrated in the 2024 year.

Q4: How did AEP's profitability change in 2024 and what were the main drivers?
A4: Profitability improved notably in 2024 versus 2023, driven mainly by favorable IRS PLRs treating NOLCs, positive rate case outcomes and higher transmission and sales volumes; these gains were partly offset by CCR-related costs, severance and a Texas refund provision. The change was material for the year 2024 compared to 2023.

Q5: What trend is described regarding regulatory scrutiny and proceedings of AEP's utility operations?
A5: Regulatory scrutiny and formal proceedings have increased or remained active in recent years, with regulators probing prudence of costs, past capital expenditures and rate filings, raising the risk of disallowances or adjustments (moderate to significant regulatory pressure).

Q6: How is AEP's transmission business trending and what major initiatives are underway?
A6: AEP's transmission business is growing materially, driven by substantial multibillion-dollar investments through State Transcos and transmission joint ventures; management has been accelerating construction and filing forward-looking formula rates, with significant projects planned through the next several years.

Q7: How did profitability margins trend during the recent period?
A7: Profitability margins improved in 2024 (notably higher net income attributable to common shareholders year-over-year) due to one-time tax/regulatory benefits and operational revenue gains, although margins faced pressure from increased operating costs like CCR compliance and severance charges.

Q8: How have regulatory and tax accounting developments affected AEP's balance sheet in the recent period?
A8: Regulatory and tax changes materially reshaped the balance sheet in 2024 — AEP recognized regulatory assets for NOLC revenue requirements, remeasured Excess ADIT (reducing regulatory liabilities), and recorded related deferred tax impacts, producing noticeable shifts in deferred tax and regulatory account balances. These were significant in 2024 as the PLRs and rate filings were implemented.

Q9: How does regulatory lag in rate cases affect AEP's earnings trend?
A9: Regulatory lag from lengthy base rate proceedings tends to depress or delay recovery of costs, causing earnings to be lower or more volatile in the near term (noticeable impact over the most recent rate cycles).

Q10: What is the role and recent trend of the Generation & Marketing segment?
A10: The Generation & Marketing segment operates competitive retail supply and wholesale trading businesses that face market-driven volatility; it remains an important but more market-exposed part of AEP's portfolio, and the company has recently narrowed the segment through asset sales while maintaining a sizable retail customer base.

*Figure A7.* Checklist Example in the 10-K Scenario.

---

**Checklist Example in the GLOBEM Scenario**

Q1: How did the user's emotional support given change during the experiment?
A1: It improved.

Q2: How did the user's emotional support received change during the experiment?
A2: It improved.

Q3: How did the user's depressive symptoms change during the experiment?
A3: It worsened.

Q4: How did the user's anxiety levels change during the experiment?
A4: It worsened.

Q5: How did the user's social adaptation change during the experiment?
A5: It remained almost the same.

---

*Figure A8.* Checklist Example in the GLOBEM Scenario.

and MIMIC induce more interaction rounds than signal-based datasets such as GLOBEM, and the resulting distributions are also more uniform.

## J. Valid Insight Ratio

When generating message-wise insights, the LLM is prompted to decide autonomously whether the outcome of the current interaction is meaningful, as shown in Figure A15. If it is not, the model outputs "NO INSIGHT". An interaction is treated as uninformative and marked as "NO INSIGHT" only when the content is:

1. insufficient to support any insight

2. failed function call

3. when the interaction only invokes descriptive tools such as `list_files`, `describe_table`, `get_database_info`, or `get_field_description`

The proportion of meaningful insights is then computed over all generated message-wise insights. In practice, at least two to three interactions are expected to involve descriptive tool calls and therefore produce no insight, which implies that the effective insight ratio cannot reach 100%.

As shown in Figure A12, the height of each bar indicates the total number of message-wise insights generated, while the solid segment represents the number of valid insights. Stronger LLMs, including Claude 4.5 Sonnet, DeepSeek, and GLM, sustain a high effective insight ratio while generating a large volume of insights, which denotes that these models maintain high information density during the exploration.

## K. Tool Execution Time Distribution

As shown in Figure A13, the tool execution times of different models vary across scenarios. The duration of tool calls partially reflects the complexity of tool usage by each LLM. We report only the tool execution time, as the total runtime, including LLM generation, is affected by multiple factors such as API service stability and local GPU performance. Presenting the tool execution time allows for a fair comparison across different LLMs.

In the MIMIC scenario, GPT-5-mini and Qwen3-30B-A3B exhibit significantly longer tool execution times than other models, yet their overall performance does not surpass them. This indicates that complex queries do not necessarily yield more meaningful insights. Efficient and precise database interaction will be a crucial metric for evaluating a model's data

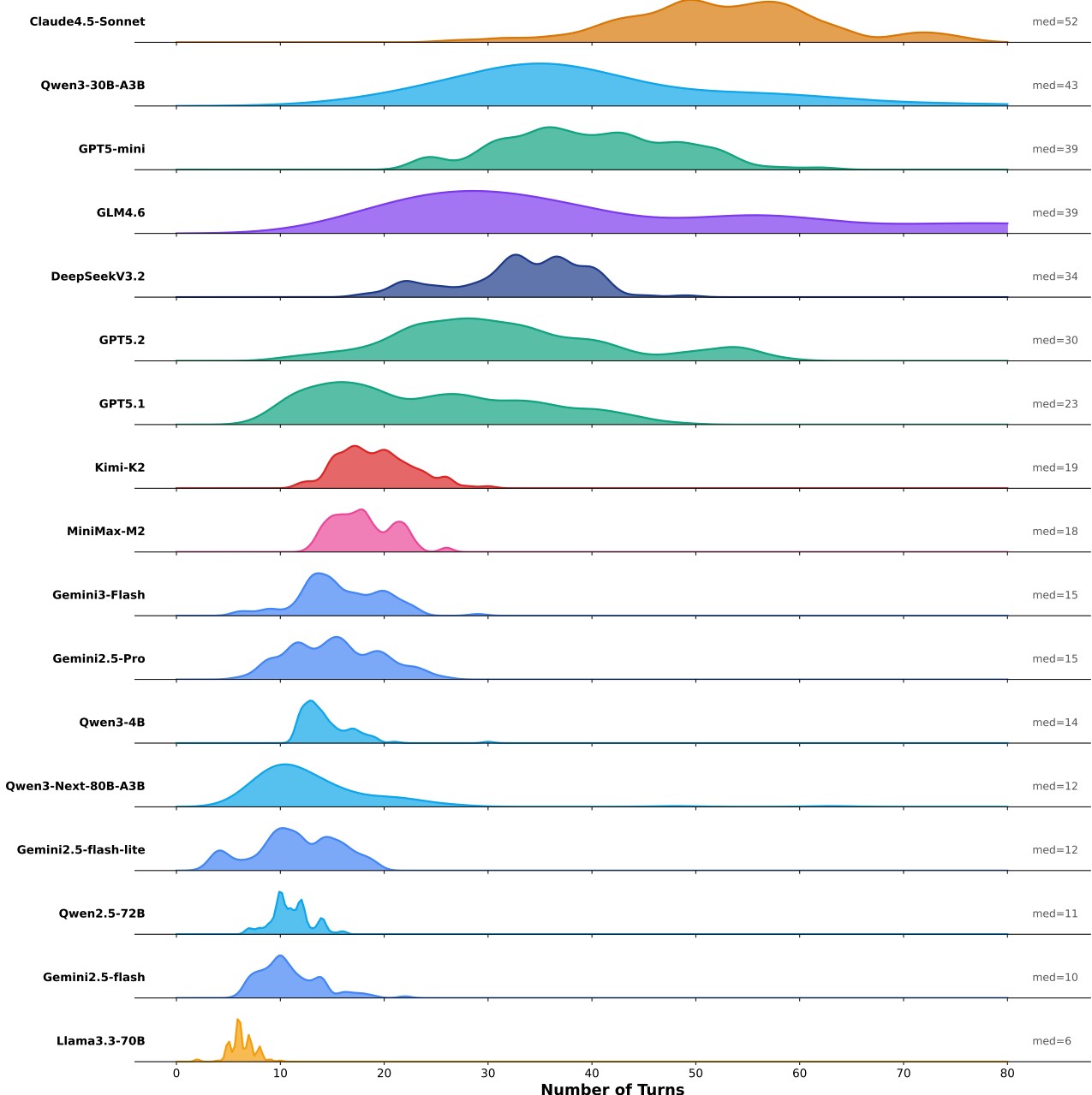

*Figure A9.* Interaction turn distribution on MIMIC scenario.

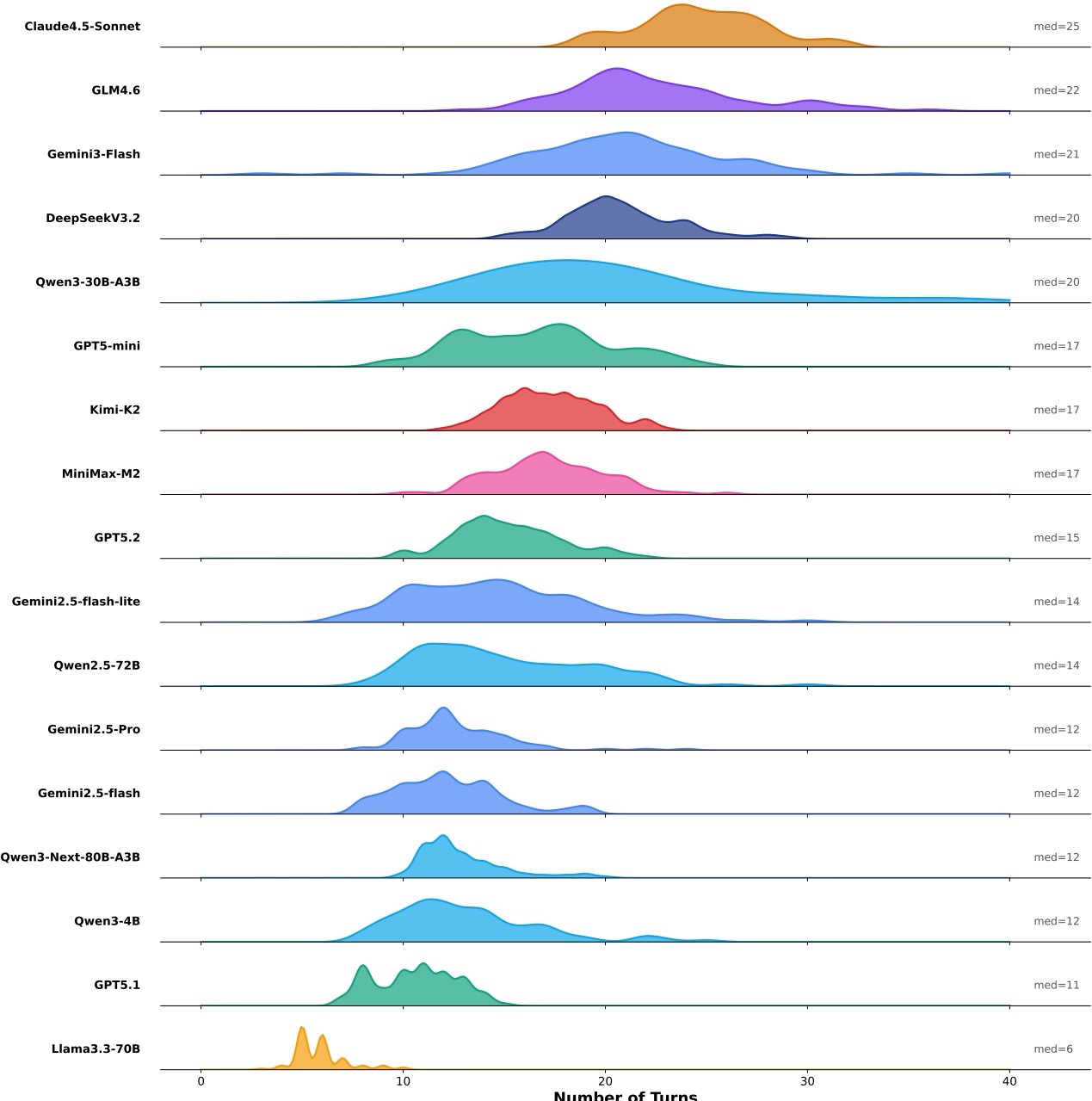

*Figure A10.* Interaction turn distribution on GLOBEM scenario.

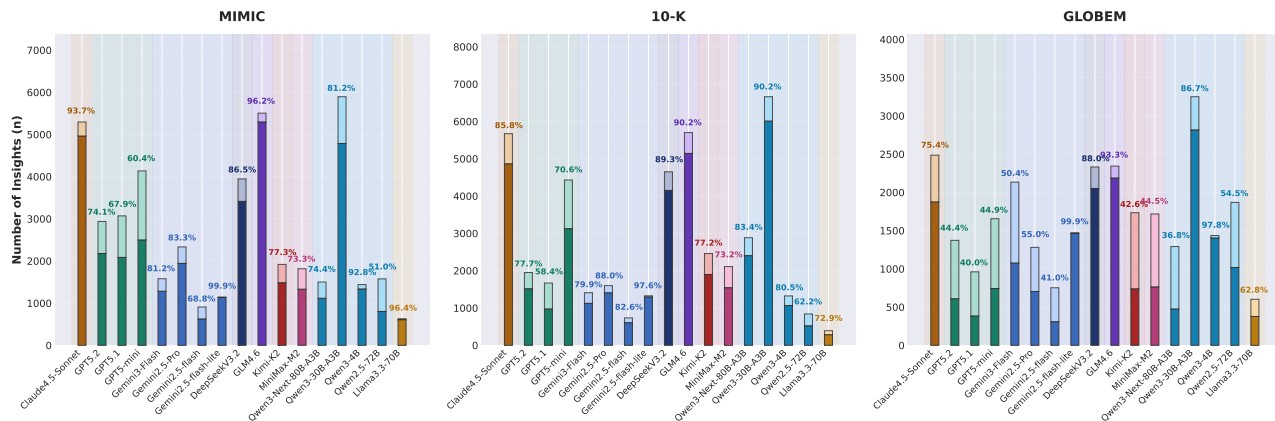

*Figure A11.* Interaction turn distribution on 10-K scenario.

*Figure A12.* Valid insight ratio distribution.

insight capabilities, encompassing high query efficiency (useful information returned per unit execution time), extraction of more information in fewer interaction rounds, and derivation of more meaningful insights from the same data.

## L. Prompts

### L.1. System Prompt

Figure A14 shows the system prompt used in DDR-Bench. It mainly enforces a ReAct-style multi-turn interaction and specifies strict requirements on the output format. Some constraints are repeatedly emphasised in uppercase, as smaller models otherwise struggle to follow the instructions and conduct reasonable multi-step agentic exploration. For larger models, the system prompt could be further simplified; however, for a fair comparison, we apply the same system prompt to all models.

### L.2. Message-wise Insight Generation Prompt

Figure A15 shows the prompt used to generate message-wise insights. This prompt is independent from the agent trajectory: message-wise insight generation is not part of the agent's ReAct trace. Instead, at each ReAct turn, we invoke this prompt to produce an insight and then continue the trajectory, while all calls are made to the same underlying model. The model only observes the agent's ReAct content from the current turn and the corresponding environment execution results. Since we organise the multi-turn interaction between the agent and the environment in a dialogue format, the agent corresponds to the assistant and the environment to the user, which explains the use of the placeholders `assistant_content` and `user_content`.

### L.3. Novelty Pairwise Judge Prompts

Figure A16, A17, and A18 provide the prompt for the pairwise novelty judge in the scenario of MIMIC, 10-K and GLOBEM, respectively.

### L.4. LLM-as-a-Checker Prompts

Across the three scenarios, the LLM is prompted to act as a checker according to the same principle, namely, to assess whether the provided insights are sufficient to support answering the questions specified in the checklist items, including

1. Determine if the messages can provide evidence to support the answer

2. Identify which specific message(s) by their index numbers `[Message X]` support or contradict the answer

3. Extract the evidence text from the relevant message(s)

4. Classify the context quality into one of three categories: `CORRECT_INFO`: Messages contain information that serves as evidence or support for the answer; `INCORRECT_INFO`: Messages contain information that contradicts the answer; `INSUFFICIENT_INFO`: Messages lack sufficient information to answer the question.

The proportion of `CORRECT_INFO` is calculated as the final accuracy. In addition, the GLOBEM scenario uses a closed-form checklist and therefore does not require the LLM for checking. Instead, the LLM directly answers the questions based on the insights, and accuracy is computed by exact match against the ground truth. The reason that GLOBEM and 10 K do not adopt a sampling-based checking procedure is that answers to open-ended questions are difficult to normalise.

Furthermore, the prompt includes scenario-specific evaluation criteria as well as JSON return fields to facilitate downstream data processing. Details can be found in the supplementary code at `evaluate/prompts.py`.

## M. Trajectory Sample

Figure A19, A21, and A20 provide a trajectory sample in the scenario of MIMIC, 10-K and GLOBEM, respectively. The trajectory includes the reasoning, action, observation (execution results from the database environment) and two kinds of insights, message-wise insight $I_m$ and trajectory-wise insight $I_t$. We anonymise the task entity id in the MIMIC and GLOBEM.

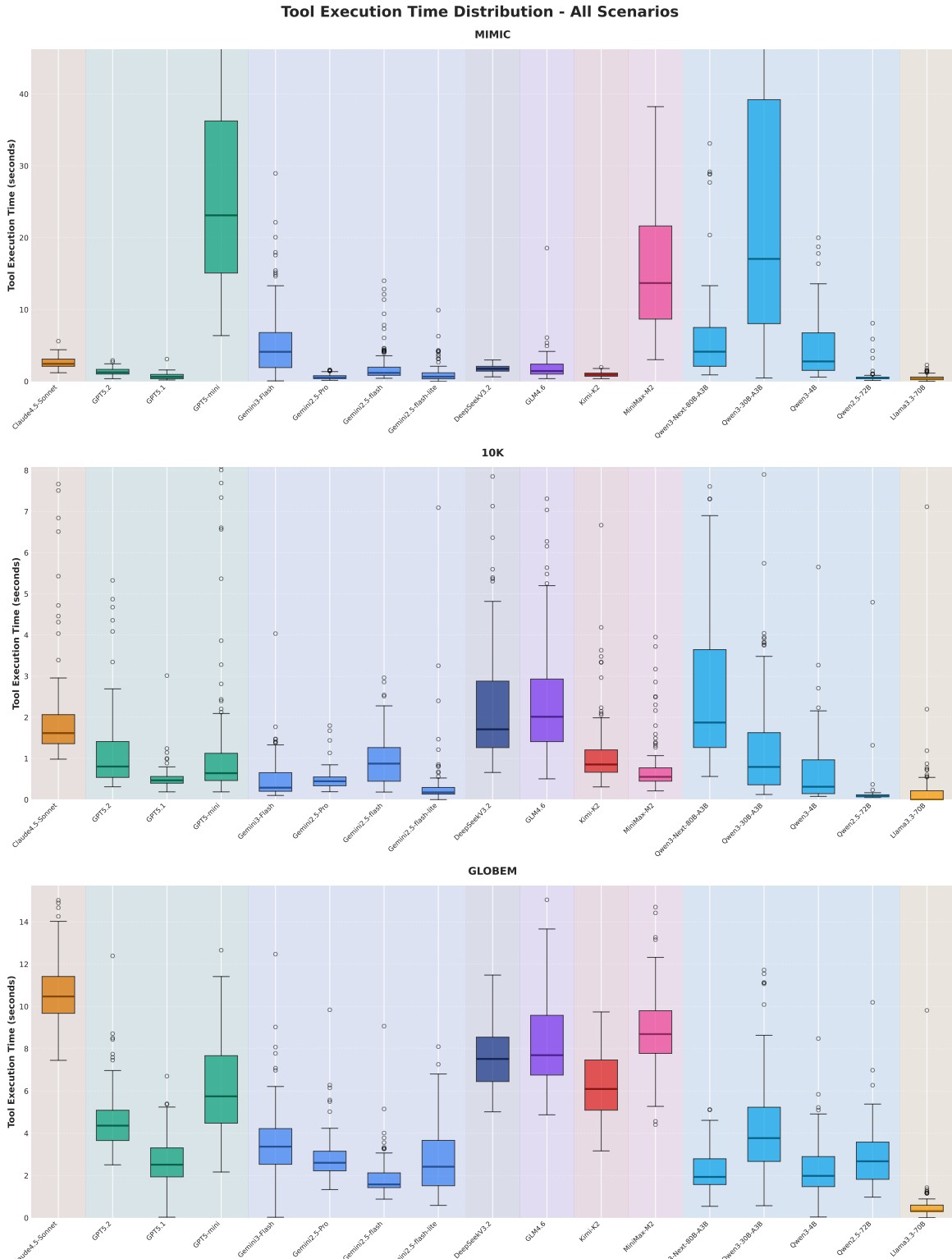

*Figure A13.* Average tool execution time per trajectory.

---

**System Prompt in DDR-Bench**

You are an autonomous data analysis agent. Your task is to keep exploring and analyzing data for the given task.

IMPORTANT INSTRUCTIONS:
1. always respond in a ReAct style: return what you're thinking and planning to do, and then call the appropriate tool. RETURN BOTH THE TEXT CONTENT AND THE TOOL CALL.
2. you can only call one tool every turn.
3. your reasoning should contain insights derived from last turn's tool call results. BUT DO NOT INCLUDE ANY INSIGHTS OR REASONING IN THE TOOL CALLS. TOOLS ARE ONLY FOR DATA EXPLORE.
4. you should try your best to use the tools to get more information. keep exploring, build more and more complex params as turns go on and you will discover more in the data.
5. first use the tools to check what data is available to you.

TASK COMPLETION: When you can not gather more information, send a message that starts with "FINISH:" followed by your all insights collected from the whole dialogue and tool calls.
Only use "FINISH:" when you are absolutely certain that no more information can be gathered.
Carefully use "FINISH:" in your message since it will immediately end the session. Think twice before using it.

YOUR TASK: `{task}`

Analyze the task and use the available tools to accomplish it step by step.

ALWAYS RETURN in the format of text content with a tool call, no just return the tool call.

*Figure A14.* System prompt in DDR-Bench. The system prompt gives a detailed guideline to ensure that the LLMs follow the ReAct style for exploration, but it does not mention any specific question, task or goals. The task placeholder in this prompt denotes the simple start prompt part, like "start analysis $\{entity\_id\}$".

---

**Prompt for Generating Message-wise Insight $I_m$ in DDR-Bench**

Based on the following tool execution, provide a brief insight about what was discovered or learned:

The reason and action to use the tool:
`{assistant_content}`

Tool execution result:
`{user_content}`

Provide a concise insight (1-3 sentences) about what this reveals:
1. It has to be related to the task: task.
2. If there is no insight or error in the tool execution, respond with 'NO INSIGHT'.
3. If it only use the data description tools (e.g. tools like `list_files`, `describe_table`, `get_database_info`, `get_field_description`), respond with 'NO INSIGHT'.
4. The insight from data should answer the question raised in the reason to execute this tool. Focus on this point.
5. Keep all the data or statistics needed in your generated insight. ONLY respond with the insight.

*Figure A15.* Prompt for Generating Message-wise Insight $I_m$ in DDR-Bench.

Novelty Pairwise Judge Prompt for the MIMIC Scenario

You are an expert clinical evaluator. You will compare two sets of novel clinical insights about the same patient.

Patient Context: These insights were generated during clinical data analysis but were NOT used to answer specific clinical questions correctly. They represent potentially valuable but unused observations.

Your task: Determine which set provides MORE clinically actionable and useful information for understanding the patient's condition.

Consider:
- Diagnostic value: Does it help identify or clarify diagnoses?
- Treatment implications: Does it inform treatment decisions?
- Prognostic relevance: Does it help predict outcomes?
- Clinical actionability: Can clinicians act on this information?
- Depth of insight: Does it reveal meaningful patterns or connections?
- Do not be biased by the length, number of insights, fluency, etc. Just focus on the usefulness of the insights.

Insights from Model A:
{insights_a}
Insights from Model B:
{insights_b}

Respond in EXACTLY this format (two lines):
Line 1: One sentence explaining your reasoning (max 100 words)
Line 2: Your decision - ONLY one of: MODEL_A, MODEL_B, or TIE

Example: Model A provides more specific diagnostic details about the patient's cardiovascular condition and treatment implications.
MODEL_A

Your response:

*Figure A16.* Novelty Pairwise Judge Prompt for the MIMIC Scenario.

**Novelty Pairwise Judge Prompt for the 10-K Scenario**

You are an expert financial analyst. You will compare two sets of novel financial insights about the same company.

Company Context: These insights were generated during 10-K analysis but were NOT used to answer specific financial questions correctly. They represent potentially valuable but unused observations.

Your task: Determine which set provides MORE valuable information for investment analysis and business understanding.

Consider:
- Investment value: Does it inform investment decisions?
- Business strategy implications: Does it reveal strategic directions or challenges?
- Financial health indicators: Does it highlight financial strengths or risks?
- Competitive positioning: Does it clarify market position or advantages?
- Depth of insight: Does it reveal meaningful patterns or trends?
- Do not be biased by the length, number of insights, fluency, etc. Just focus on the usefulness of the insights.

Insights from Model A:
{insights_a}
Insights from Model B:
{insights_b}

Respond in EXACTLY this format (two lines):
Line 1: One sentence explaining your reasoning (max 100 words)
Line 2: Your decision - ONLY one of: MODEL_A, MODEL_B, or TIE

Example: Model A provides more specific financial metrics and strategic insights that are more actionable for investment decisions.
MODEL_A

Your response:

*Figure A17.* Novelty Pairwise Judge Prompt for the 10-K Scenario.

**Novelty Pairwise Judge Prompt for the GLOBEM Scenario**

You are an expert behavioral psychologist and mental health researcher. You will compare two sets of novel behavioral insights about the same user.

User Context: These insights were generated during passive sensing data analysis but were NOT used to answer specific mental health questions correctly. They represent potentially valuable but unused observations.

Your task: Determine which set provides MORE useful understanding of their mental health patterns and behavioral trends.

Consider:
- Behavioral indicators: Does it identify meaningful behavior patterns?
- Mental health relevance: Does it relate to psychological well-being?
- Longitudinal trends: Does it reveal changes over time?
- Actionability: Can this inform interventions or support?
- Depth of insight: Does it reveal meaningful connections in the data?
- Do not be biased by the length, number of insights, fluency, etc. Just focus on the usefulness of the insights.

Insights from Model A:
{insights_a}
Insights from Model B:
{insights_b}

Respond in EXACTLY this format (two lines):
Line 1: One sentence explaining your reasoning (max 100 words)
Line 2: Your decision - ONLY one of: MODEL_A, MODEL_B, or TIE

Example: Model A identifies more specific behavioural patterns related to mental health that could inform targeted interventions.
MODEL_A

Your response:

*Figure A18.* Novelty Pairwise Judge Prompt for the GLOBEM Scenario.

---

**Trajectory Sample on MIMIC**

🤖 Agent: I'll help you analyze patient xxxxxx. Let me start by exploring what data is available in the database.
```
execute: {"tool": "get_database_info", "arguments": {}}
```

▶ Environment: {'database_path': 'dataset/mimic.db', 'table_count': 27, 'tables':['hosp
_admissions', 'hosp_d_hcpcs'.....], 'tables_info': [{'name': 'hosp_admissions',
'comment': "Hospital admissions for each patient. Defines......

🤖 Agent: Good! I can see this is a MIMIC database with 27 tables containing patient medical data. Now let me search for patient xxxxxx starting with the patient demographics table.
```
execute: {"tool": "execute_query", "arguments": {"query": "SELECT * FROM hosp_patients
 WHERE subject_id = xxxxxx"}
```

▶ Environment: {'cols': ['subject_id', 'gender', 'anchor_age', 'anchor_year', 'anchor
_year_group', 'dod'], 'data': [['xxxxxx', 'M', '74', '2182', '2008 - 2010',
'2185-06-15']], 'count': 1}

💡 Agent Message-wise Insight: Patient xxxxxx is a 74-year-old male with an anchor year of 2182 (corresponding to the real-world period 2008-2010) who died on June 15, 2185, indicating he survived approximately 3 years beyond the anchor year recorded in this database.

*...(46 interactions)*

🤖 Agent: No prior TIA documented. Let me check CHA2DS2-VASc risk factors systematically for stroke risk with AFib.
```
execute: {"tool": "execute_query", "arguments": {"query": "SELECT \n  'Has CHF' as
risk_factor, COUNT(DISTINCT d.hadm_id) as admissions_with_dx\nFROM hosp_diagnoses_icd
d\nWHERE d.subject_id = xxxxxx AND d.icd_code LIKE '428%'\nUNION ALL\nSELECT 'Has HTN',
COUNT(DISTINCT d.hadm_id)\nFROM hosp_diagnoses_icd d\nWHERE d.subject_id =......
```

▶ Environment: {'cols': ['risk_factor', 'admissions_with_dx'], 'data': [['Has CHF', '15'],
['Has HTN', '18'], ['Has DM', '17'], ['Has Vascular Dz', '9'], ['Has AFib', '16']],
'count': 5}

💡 Agent Message-wise Insight: This patient (ID: xxxxxx) has documented atrial fibrillation and multiple CHA2DS2-VASc risk factors present across their 18 admissions, including hypertension (18 admissions), diabetes (17 admissions), CHF (15 admissions), and vascular disease (9 admissions), indicating a high stroke risk profile that would warrant anticoagulation therapy consideration.

*...(12 interactions)*

💡 Agent Trajectory-wise Insight: Excellent! I now have a comprehensive understanding of Patient xxxxxx. Let me compile all the insights. FINISH:

```
## COMPREHENSIVE ANALYSIS OF PATIENT xxxxxx

### DEMOGRAPHIC PROFILE
- **Patient ID**: xxxxxx
- **Gender**: Male
- **Age at anchor**: 74 years (anchor year 2182)
- **Age at death**: 77 years
- **Date of death**: June 15, 2185
......

This patient exemplifies the challenges of managing elderly patients with multiple,

severe chronic conditions and the high mortality risk associated with acute stroke

in the setting of advanced multi-organ disease.
```

*Figure A19.* Trajectory Sample on MIMIC

---

### Trajectory Sample on GLOBEM

🤖 Agent: I'll help you analyze how user behavior patterns change over time for user xxxxxx...
execute: {"tool": "list_files", "arguments": {}}

```
Environment: {'files': [{'name': 'activity_allday_raw.csv', 'path': '
activity_allday_raw.csv', 'type': 'file', 'size': 2067543, 'modified':
'2025-11-21T20:08:33', 'description': 'Daily activity metrics including movement and
exercise patterns.'}......], 'count': 6, 'pattern': None, 'recursive': True}
```

*...(19 interactions)*

🤖 Agent: Excellent! I can see a clear behavioral shift starting April 30th. Let me now examine circadian rhythm patterns and correlate multiple behavioral dimensions.
execute:

```python
import pandas as pd
import numpy as np
user_pid = 'xxxxxx'
# Merge multiple data sources for correlation analysis
phone_df = pd.read_csv('phone_usage_allday_raw.csv')
loc_df = pd.read_csv('location_allday_raw.csv')
conn_df = pd.read_csv('connectivity_allday_raw.csv')
# ...(66 lines)
for label, (metric, before_val, after_val) in metrics.items():
    if pd.notna(before_val) and pd.notna(after_val) and before_val != 0:
        change = ((after_val - before_val) / before_val * 100)
        print(f\"{label:<30}:{change:>+7.1f}%\")
```

```
Environment: Critical Behavioral Shift: Day 28 (April 30, 2018) BEFORE SHIFT
 (Days 1-27):
 Phone unlocks:        34.3 per day
 Screen time:          87.7 minutes
 ......
 Distance traveled                :  +211.2\%
 Location transitions             :   +66.4\%
 Social proximity scans           :   +51.6\%
```

💡 Agent Message-wise Insight: A dramatic behavioral shift occurred on Day 28 (April 30, 2018) for user xxxxxx, indicating a potential psychological status change. Phone unlocks increased 197.7% (34.3→102.1/day), screen time rose 134% (87.7→205.3 min/day), first phone use shifted 77.4% earlier (434→98 min after midnight), distance traveled surged 211.2% ......

*...(6 interactions)*

💡 Agent Trajectory-wise Insight: Excellent! I have completed a comprehensive analysis. I have gathered extensive information about the user's behavioral patterns and their changes over time. Let me now provide a final comprehensive summary. FINISH:

```
# COMPREHENSIVE BEHAVIORAL ANALYSIS: USER xxxxxx

## Overview
I analyzed 92 days of multimodal behavioral data (April 3 - July 3, 2018) for

user xxxxxx, examining patterns across 6 behavioral domains......User xxxxxx exhibited

a **statistically significant, clinically meaningful, multi-dimensional behavioral

transformation** starting precisely on Day 28 (April 30, 2018). The pattern...
```

*Figure A20.* Trajectory Sample on GLOBEM

---

**Trajectory Sample on 10-K**

🤖 Agent: I'll analyze the structured financial data from Phillips 66 (CIK: 1534701). Let me start by exploring what data is available in the database.
execute: {'database_path': 'dataset/10-k.db', 'table_count': 4, 'tables': ['companies', 'company_addresses', 'company_tickers', 'financial_facts'..., 'filtered_out_count': 5}}

*...(8 interactions)*

🤖 Agent: Perfect! Now I can see the pattern. For fiscal year 2024, the end_date is 2024-12-31, and the revenue is $143.153 billion. The highest value shown (like $169.99B for end_date 2022-12-31) are historical comparatives. Let me now get a clean view of annual revenues and net income by matching fiscal_year to end_date year.
execute: {"tool": "execute_query", "arguments": {"query": "SELECT fiscal_year, fact_name, fact_value, end_date FROM financial_facts WHERE cik = '1534701' AND fiscal_period = 'FY' AND fact_name IN ('RevenueFromContractWithCustomerExcludingAssessedTax', 'NetIncomeLoss') AND CAST(strftime('%Y', end_date) AS INTEGER) = fiscal_year ORDER BY fiscal_year DESC, fact_name", "limit": 50}

⌨️ Environment: {'cols': ['fiscal_year', 'fact_name', 'fact_value', 'end_date'], 'data': [['2024', 'NetIncomeLoss', '2117000000.0', '2024-12-31'], ['2024', 'RevenueFromContractWithCustomerExcludingAssessedTax', '143153000000.0', '2024-12-31'], ['2023', 'NetIncomeLoss', '7015000000.0', '2023-12-31'], ['2023', 'RevenueFromContractWithCustomerExcludingAssessedTax', '147399000000.0', '2023-12-31'] ...], 'count': 50}

💡 Agent Message-wise Insight: Phillips 66 experienced significant revenue volatility over the past five years, with revenues peaking at $170.0 billion in 2022 before declining 15.8% to $143.2 billion in 2024. More concerning is the sharp decline...

*...(70 interactions)*

💡 Agent Trajectory-wise Insight: Perfect! Now I have a complete picture of Phillips 66's financial data. I have gathered extensive information covering: - Income statement data (2012-2024) - Balance sheet data (2012-2024) - Cash flow data (2012-2024) - Quarterly trends (2023-2025 Q2) - Key metrics like EPS, dividends, share repurchases, debt, working capital, etc. Now I can provide comprehensive insights based on this data. FINISH: Based on my comprehensive analysis of Phillips 66's (CIK: 1534701) structured financial data from 2012-2025 (through Q2 2025), here are the material quantitative patterns, trends, and relationships that investors should consider:

```
# REVENUE AND PROFITABILITY TRENDS

**Revenue Volatility Pattern (2018-2024):**
- Peak: $169.99B in 2022 (+52.5% from 2021's $111.48B)
- Decline: $147.40B in 2023 (-13.3% YoY)
- Further decline: $143.15B in 2024 (-2.9% YoY)
- This reveals significant sensitivity to commodity price cycles, with 2022 benefiting
from energy price spikes post-pandemic

**Net Income Cyclicality:**
- 2020: -$539M (loss year during pandemic)
- 2021: $1.32B (recovery but weak)

......

The data reveals Phillips 66 is in a challenged operating environment with compressed
margins, declining profitability, and aggressive capital returns that may not be
sustainable at current earnings levels. The company is drawing on balance sheet
strength built during 2022's exceptional year, but with declining cash, rising
leverage, and deteriorating coverage ratios, financial flexibility is eroding.
Investors should assess whether the current 27.8x P/E ratio appropriately reflects both
cyclical risks and the company's strategic positioning in a potentially structurally
challenged refining industry.
```

*Figure A21.* Trajectory Sample on 10-K

