# OpenReview forum: "Hunt Instead of Wait: Evaluating Deep Data Research on Large Language Models"
_ICML.cc/2026/Conference — ICML 2026 regular_

### Official Review · Reviewer_DCkF · 2026-03-06

**Soundness:** 3
**Presentation:** 3
**Significance:** 3
**Originality:** 3
**Overall Recommendation:** 5
**Confidence:** 3

**Summary:**

This paper introduces Deep Data Research, a new task formulation designed to evaluate the investigatory intelligence of agents. LLM agents are given only a structured database and a minimal start prompt and are expected to autonomously explore the data, form and test hypotheses, decide when to stop, and produce a report of insights. To evaluate this, the authors build DDR-Bench, and benchmark 20+ LLMs and follow up with rich analyses of scaling behavior, exploration patterns, self-termination, and module ablations.

**Compliance With Llm Reviewing Policy:**

Affirmed.

**Key Questions For Authors:**

1. Checklist construction quality: How many candidate checklist items were generated from the unstructured components before expert screening, and what was the filtering rate? Was inter-annotator agreement measured during the expert screening phase? What if a checklist item turned out to be very difficult to infer from the structured data alone in practice?

2. How does the benchmark handle "redundant" or "brute-force" exploration? Is there a cost or efficiency metric included in the final score?

**Limitations:**

The authors discuss failure modes and hallucination risks. I would suggest adding some more thoughts on the potential environmental/monetary cost of running such long-horizon benchmarks (and whether the authors expect academic labs would be able to handle it), as well as the risk of "over-fitting" to the specific style of GPT-generated checklists.

**Strengths And Weaknesses:**

Strengths:
- I like the distinction the paper makes between executional intelligence and investigatory intelligence.
- I like the evaluation strategy. The way authors use unstructured text to generate checklists for structured data exploration is a very smart workaround for the "open-ended evaluation" problem. It makes the assessment objective and scalable. I also like the novelty analysis & the benchmark itself is an interesting contribution!
- I like the analyses on interaction patterns and cost. The failure mode taxnomy helps ground future work in this direction.

Weakness
- I would like to learn more about the checklist practicality. By construction, the checklist is derived from the unstructured text component of the database. But the LLM agents are exploring the structured component. There's an implicit assumption that structured data exploration will surface the same insights that appear in the clinical notes / 10-K text sections. This seems reasonable for MIMIC and 10-K (where the text closely reflects the data), but I'd like to see more explicit discussion of cases where the structured data genuinely cannot support a checklist item. The authors mention human experts confirm "surjective" but what about items that technically could be inferred but require extremely complex multi-hop queries that might be less reasonable to expect or less important to discover?
- I wonder how closely the benchmark setup matches real-world analyst workflows. For example, human analysts often rely heavily on visualizations and iterative questioning with collaborators, which are not modeled here.
- I wonder if a model could "win" this benchmark by just brute-force querying every table and field until it hits the checklist items, rather than being "intelligent" about its investigation. How does the benchmark penalize inefficient exploration or redundant queries?

---

> ### Author Rebuttal · Authors · 2026-03-29
>
> Thank you for the recognition and helpful suggestions! We hope the following clarifications help resolve your concerns.
>
> ### Q1: Checklist Practicality
> - This is an excellent point and an open question worth discussing.
>   -  As a benchmark, its verifiability comes from the Checklist. Items can be inferred directly or indirectly from structured data. For those insights that are not covered by the Checklist but discoverable from structured data, please see Section 3.2 on *Novelty*. Currently, our automated evaluation uses pairwise comparison. Absolute evaluation of novel insights may still require human judgment or a rubric-based approach.
>   -  Another case is that some checklist items cannot be supported by structured data. We confirmed this via human expert annotations. Only items with agreement from both experts were retained. From the initial 2×100×10 items (2 scenarios, 100 entities, 10 candidate items per entity), 83.6% were retained after agreement filtering, and 81.15% remained after further quality control. In our observations, unsupported items typically arise not from complexity or being less important, but due to two reasons:
>         - The item cannot be inferred solely from structured data and requires additional databases.
>         - The item could be inferred indirectly, but cannot be confirmed without extra context. For example, in 10-K, financial health can be inferred from statements, but the specific reasons (e.g., overexpansion or failed acquisitions) are missing.
>   - It remains possible that checklist items are not the most salient insights in the database. We did not attempt to define “most valuable insights”; instead, we collected reasonable data. In high-stakes scenarios, unstructured text is typically the most important source. For other datasets, human annotation may be necessary.
> ---
> ### Q2: Workflow Differences
> - We agree with you. DDR-Bench is a clean starting point. We plan to introduce multi-modalities and multi-agent behaviours in the future.
> - But in some cases, models do not necessarily process data like humans. While human engineers rely on visualisation, models can detect trends and distributions directly from tables.
> - The benchmark is designed to be *in the wild*. We do not explicitly model analysts’ workflows; instead, we ensure authenticity and alignment by sourcing data from real-world analysis outputs.
> ---
> ### Q3: Win by Brute-Force Retrieval
> -  This benchmark is designed to *Go Beyond Information Retrieval*. Many items require complex computations and reasoning. For GLOBEM, retrieval alone cannot solve the task because structured data (wearable data) and checklist items (mental status) are from different domains.  For 10-K and MIMIC, please see the cases below:
>    - MIMIC, for patient 10670364: “What induction chemotherapy regimen did the patient receive?” Answer: 7+3 induction. The term “7+3” does not exist in any table or field; it is a clinical oncology term for “7 days cytarabine + 3 days anthracycline.” The agent must infer this from raw prescription data:
>      - Identify AML diagnosis from `diagnoses_icd` (ICD 20500/20501).
>      - Search prescriptions for chemotherapy drugs.
>      - Detect two dose levels of Cytarabine (190mg ×1 vs 5580–5730mg ×14).
>      - Check first admission: Cytarabine 190mg + Daunorubicin 170mg started the same day.
>      - Combine all these findings → 7+3 induction chemotherapy.
>    - 10-K: for DHR (Danaher, CIK 313616): “How did the company’s profitability margins change over the period?” Answer: Operating profit margins declined materially (27.6% → 20.4%, 720bps) due to impairments, acquisition dilution, and Veralto spin-off. Reasoning:
>      - Profit margins are not direct fields; they must be computed from Revenue, OperatingIncomeLoss, NetIncomeLoss.
>      - Danaher divested the EAS business at the end of 2023; the model must identify discontinued operations ($543M) and understand its impact.
>      - Attribution requires four drivers: impairment, acquisition dilution, SG&A ratio rise, and revenue decline.
>      - Model further calculates derived metrics (Current Ratio, Quick Ratio, DSO, R&D intensity) in SQL to support judgment.
> -  Limited model context and large databases prevent brute-force retrieval.
> -  As a benchmark, we do not penalise, we only observe; metrics only record how many checklist items are discovered. Section 4.1 observes LLM behaviour from the scaling of turns, tokens and cost. Inefficient exploration shows disadvantages on scaling curves.
> ---
> ### Q4: Limitations
> - Costs shown in the paper are real, accounting for KV cache hits (~70%). Tokens from cache are usually billed at ~1/10 price, reducing cost.
> - We will note risks of using GPT as a checklist generator.
> - Thank you for your suggestions. We will detail all these concerns in the limitations section.

---

> > ### Author Rebuttal · Reviewer_DCkF · 2026-04-02
> >
> > Thank you for your detailed discussions! I don't have more questions. My current score already reflects my assessment and I will keep it.

---

> > > ### Author Response · Authors · 2026-04-02
> > >
> > > Thank you for your recognition! We will incorporate a more detailed discussion into the limitations section, as you suggested above.

---

### Official Review · Reviewer_9AgP · 2026-03-11

**Soundness:** 3
**Presentation:** 3
**Significance:** 2
**Originality:** 2
**Overall Recommendation:** 4
**Confidence:** 4

**Summary:**

The paper presents the concept of investigatory intelligence which is the ability of LLMs to mimic real-world data analysis by starting from raw data and conducting open-ended explorations, rather than only answering predefined data questins. The authors introduce Deep Data Research (DDR) and construct DDR-Bench, a checklist-based benchmark designed to evaluate an agent’s ability to decide what questions to ask and explore in the first place. Their results show that current LLM agents still struggle with long-horizon exploration and autonomous investigation.

**Compliance With Llm Reviewing Policy:**

Affirmed.

**Final Justification:**

The authors discuss a central topic around evaluating investigatory intelligence in agentic LLMs, and the rebuttal provides helpful additional evidence and clarifications.

The new experiments comparing different agent designs strengthen the claim that the observed limitations are not primarily due to the minimal scaffold, and the added details on checklist construction and validation improve confidence in the evaluation pipeline. This paper studies an important concept and the empirical analysis is solid.

Overall, my main concerns have been addressed, and I increase my confidence in the assessment.

**Key Questions For Authors:**

1. Since the agent scaffold is intentionally minimal, did the authors explore whether stronger agent designs (e.g., some planning modules, memory mechanisms, or multi-agent frameworks) could significantly change performance on DDR-Bench? This would help clarify whether the observed limitations are due to LLM capability or agent design.
2. How did the authors ensure that the checklist sufficiently covers meaningful insights that could be discovered from the database? Have the authors evaluated inter-annotator agreement or used other validation measures to assess checklist reliability?
3. This DDR-Bench still focuses on structured database exploration. How well the authors think the DDR framework could generalize to other forms of data (unstructured data or some other open-ended research tasks)?

**Limitations:**

yes

**Strengths And Weaknesses:**

## Strengths:
- This paper introduces this new concept of investigatory intelligence, which focuses on the ability of LLM agents to conduct open-ended analysis starting from raw data rather than only answering predefined questions. This is different from many existing QA-style benchmarks for DS agents. The proposed DDR setting evaluates open-ended investigation, and the checklist-based insight verification provides an alternative to commonly used LLM-judge scoring. This framing opens an interesting direction for evaluating agentic models and highlights that current models still struggle with information gaps and autonomous investigation.
- **Clarity** The paper is clear and easy to follow. The overall narrative is well structured, and the tables and visualizations are straightforward and informative.

## Weaknesses:
- **Minimal agent baseline** I understand that the agent framework is deliberately minimal to test the intrinsic capacity of the LLMs. However, the fact that the agent scaffold being simple and no alternative agent designs being explored makes it hard to determine whether the poor performance stem from the simple scaffold design or from the LLMs themselves.
- **Checklist construction and validation** I like the idea of using checklist-based insight verification. However, the checklist is constructed from free-form reports or insights and then screened by human experts. This process could still be subjective or incomplete. Since the evaluation heavily depends on the checklist, additional validation or analysis of its coverage and reliability would strengthen the benchmark.
- **Limited methodological novelty** : The main novelty of the work lies in the conceptual framing and benchmark design. Many components build on existing work, such as a minimal ReAct-style agent and standard tool usage (SQL/Python).

---

> ### Author Rebuttal · Authors · 2026-03-29
>
> Thank you for the constructive comments. We hope the clarifications below are helpful.
>
> ### Q1: Minimal agent baseline
> - The agent is intentionally simplified to evaluate model capabilities in a clean testbed. Optimising agent scaffolding for higher scores would confound assessment.
> - DDR-Bench supports flexible tool extensions via MCP for future agent benchmarking.
> - Section 5 presents agent ablations; for example, memory modules often destabilise performance.
> - Additional experiments on Qwen3-30B-A3B, Qwen3-4B, and GPT-5-mini across Planning (Plan-and-Execute[1]), Memory (CoALA[2]), and Multi-agent (AutoGen[3]) frameworks show that complex agents mostly degrade performance compared to ReAct, except for minor planning benefits. Analysis indicates that complex frameworks affect model confidence, leading to premature self-termination or over-/under-thinking.
> ---
> | Dataset | Model | ReAct | +Plan | +Memory | +Multi-Agent |
> |---------|-------|-------|-------|--------|--------------|
> | MIMIC | Qwen3-4B | 16.67 | 8.14 | 11.46 | 4.44 |
> | MIMIC | Qwen3-30B-A3B | 20.03 | 12.27 | 13.57 | 9.04 |
> | MIMIC | GPT-5-mini | 28.81 | 23.67 | 22.22 | 12.66 |
> | 10-K | Qwen3-4B | 40.94 | 14.25 | 17.43 | 26.50 |
> | 10-K | Qwen3-30B-A3B | 42.33 | 47.59 | 37.10 | 31.80 |
> | 10-K | GPT-5-mini | 46.35 | 49.82 | 45.35 | 30.04 |
> | GLOBEM | Qwen3-4B | 26.21 | 22.76 | 23.45 | 22.30 |
> | GLOBEM | Qwen3-30B-A3B | 35.63 | 25.75 | 22.30 | 23.91 |
> | GLOBEM | GPT-5-mini | 36.09 | 28.05 | 28.72 | 25.64 |
> ---
> ### Q2: Checklist construction and validation
> - For open-ended data exploration, ideally, the database insights perfectly match our checklist, but two possibilities exist:
>   - The database contains insights not covered by the checklist. Section 3.2 (Novelty Analysis) shows pairwise comparisons of new insights; rankings based on these align closely with checklist-based rankings.
>   - The checklist contains items not directly obtainable from the database. In this case, GPT-5-mini is used to extract 10 candidate items per entity from reports and corresponding database features. Since the data are real-world and the reports align with structured databases, and LLM filtering is applied, the candidate items are of high quality. Human annotation retains only items agreed upon by both annotators, achieving 83.6% agreement. A further quality filter yields a final retention of 81.15% of items, matching the statistics reported in the paper \((774+849)/2000\).
> ---
> ### Q3: Generalisation to other tasks
> - We view this as an exciting future direction!
>   - Many open-ended research tasks (e.g., deep research) can be considered deep data research, with the database being the whole web. Core challenges remain unchanged, such as balancing exploration and improving efficiency without explicit questions (as discussed in Section 4).
>   - For unstructured data, the format may differ. We may construct concept or knowledge databases from unstructured text to enable open-ended exploration of long-text writing or comprehension.
>   - DDR-Bench emphasises not only structured database exploration, but autonomous exploration in a query-free setting. This principle applies to both structured and unstructured data and is likely essential in future research.
> ---
> Reference:
> 1. Wang, L., Xu, W., Lan, Y., Hu, Z., Lan, Y., Lee, R. K. W., & Lim, E. P. (2023, July). Plan-and-solve prompting: Improving zero-shot chain-of-thought reasoning by large language models. In Proceedings of the 61st annual meeting of the association for computational linguistics.
> 2. Sumers, T., Yao, S., Narasimhan, K. R., & Griffiths, T. L. (2023). Cognitive architectures for language agents. Transactions on Machine Learning Research.
> 3. Wu, Q., Bansal, G., Zhang, J., Wu, Y., Li, B., Zhu, E., ... & Wang, C. (2024, August). Autogen: Enabling next-gen LLM applications via multi-agent conversations. In First conference on language modeling.

---

> > ### Author Rebuttal · Reviewer_9AgP · 2026-04-04
> >
> > The rebuttal addresses my main concerns. In particular, the additional experiments comparing different agent designs provide helpful evidence that the observed limitations are not primarily due to the minimal agent scaffold. The clarification on checklist construction and validation (including inter-annotator agreement and filtering) also improves my confidence in the evaluation.
> >
> > Overall, the rebuttal strengthens the paper and resolves my key questions.

---

> > > ### Author Response · Authors · 2026-04-04
> > >
> > > We’re glad these discussions helped address your concerns, and we would appreciate it if you could consider adjusting the score based on your current evaluation of the paper. Thank you for actively engaging in the review and discussion~ :)

---

### Official Review · Reviewer_XYSz · 2026-03-12

**Soundness:** 3
**Presentation:** 3
**Significance:** 3
**Originality:** 3
**Overall Recommendation:** 5
**Confidence:** 3

**Summary:**

This paper proposes the task of Deep Data Research and the benchmark DDR-Bench to evaluate the investigatory intelligence of large language models in autonomous data exploration. Experiments on real-world datasets reveal performance differences among models and key factors in exploration efficiency. The work offers valuable insights for agent research, yet has limitations including incomplete checklist coverage, task orientation bias, and restricted exploration settings.

**Compliance With Llm Reviewing Policy:**

Affirmed.

**Final Justification:**

Most of my concerns are addressed after rebuttal, therefore I increase the score

**Key Questions For Authors:**

NA

**Strengths And Weaknesses:**

Strength:
1. It formally defines Deep Data Research and investigatory intelligence, providing a new perspective for evaluating autonomous exploration ability of large language models.
2. It constructs DDR-Bench based on three real-world datasets and uses an objective checklist-based evaluation method to reduce subjective bias.
3. It systematically analyzes agent exploration patterns and shows that effective exploration depends more on intrinsic strategies than model scale or external scaffolding.

Weakness:
1. The checklist cannot cover all potential valid insights, which may lead to false negatives in evaluation.
2. The task is closer to information retrieval rather than real data mining or scientific research, based on the cases provided in the paper.
3. The analysis of agent exploration is limited by the types of tools and the simplicity of the agent framework.
4. The benchmark only covers a small number of domains.

---

> ### Author Rebuttal · Authors · 2026-03-29
>
> We sincerely thank the reviewers and provide the following clarifications.
>
> ### Q1: Checklist Coverage
> - Exhaustively evaluating all insights in open-ended data analysis is inherently difficult. Unlike prior benchmarks relying on indirect metrics such as code correctness or visualisation richness, this work takes a concrete step toward direct evaluation.
> - Section 3.2 presents a **novelty analysis**, evaluating insights outside the checklist pair-wise to estimate false positives, showing high consistency with checklist accuracy.
> ---
> ### Q2: Similar to Retrieval
> - DDR is **designed to go beyond retrieval**. Appendix G and Figure A6 detail the required skill distribution on checklist items. Simple items (e.g., cases in Appendix A) are retrieval-like for demonstration, yet remain challenging in the query-free setting.
> - We show some complex checklist cases below for the MIMIC and 10-K databases:
>     - MIMIC (Patient 10670364): *"What induction chemotherapy regimen did the patient receive?"* → 7+3 induction. The term “7+3” does not appear in any table or field. It is a clinical oncology term referring to “7 days continuous cytarabine infusion + 3 days anthracycline.” The agent must infer the regimen from raw prescription data:
>         - From `diagnoses_icd`, identify ICD 20500/20501 → AML diagnosis.
>         - Search prescriptions for chemotherapy drugs → find Cytarabine (15 doses) + Daunorubicin (1 dose).
>         - Cytarabine presents two distinct dose levels: 190mg (1) vs 5580–5730mg (14).
>         - In the first admission, Cytarabine 190mg + Daunorubicin 170mg start on the same day.
>         - Combine all these findings → 7+3 induction chemotherapy.
>     - 10-K Dataset: For entity DHR (Danaher, CIK 313616): *“How did the company’s profitability margins change over the period?”*
> Answer: Operating profit margins declined materially (27.6% → 20.4%, 720bps), driven by impairments, acquisition dilution, and the Veralto spin-off.
> Reasoning:
>         - Margins are not stored directly. It must be computed from Revenue, OperatingIncomeLoss, NetIncomeLoss.
>         - Danaher divested the EAS business (Veralto) at end of 2023; the agent must identify discontinued operations ($543M) and account for its impact.
>         - Attribution to four drivers: impairments, acquisition dilution, SG\&A ratio increase, revenue decline.
>         - LLM also calculates derived metrics (Current Ratio, Quick Ratio, DSO, R&D intensity) in SQL to support reasoning.
> - For GLOBEM, this cross-domain dataset combines structured wearable data with checklist questions on user mental health. LLMs cannot retrieve mental state from wearable tables, and they must perform statistical analysis and reasoning.
> ---
> ### Q3:  Limited Agent Framework
> - We intentionally simplify the agent to evaluate model capabilities in a clean testbed; optimising agent scaffolding for high scores would confound assessment.
> - DDR-Bench supports flexible tool extensions via MCP for future agent benchmarking.
> - Section 5 shows agent ablations, where memory modules often destabilise performance.
> - We provide additional experiments on Qwen3-30B-A3B, Qwen3-4B, and GPT-5-mini across Planning (Plan-and-Execute[1]), Memory (CoALA[2]), and Multi-agent (AutoGen[3]) frameworks, revealing that complex agents mostly degrade performance vs ReAct, except for minor planning benefits. Analysis suggests that complex frameworks affect model confidence, leading to premature self-termination or over-/under-thinking.
>
> | Dataset | Model | ReAct | +Plan | +Memory | +Multi-Agent |
> |---------|-------|-------|-------|--------|--------------|
> | MIMIC | Qwen3-4B | 16.67 | 8.14 | 11.46 | 4.44 |
> | MIMIC | Qwen3-30B-A3B | 20.03 | 12.27 | 13.57 | 9.04 |
> | MIMIC | GPT-5-mini | 28.81 | 23.67 | 22.22 | 12.66 |
> | 10-K | Qwen3-4B | 40.94 | 14.25 | 17.43 | 26.50 |
> | 10-K | Qwen3-30B-A3B | 42.33 | 47.59 | 37.10 | 31.80 |
> | 10-K | GPT-5-mini | 46.35 | 49.82 | 45.35 | 30.04 |
> | GLOBEM | Qwen3-4B | 26.21 | 22.76 | 23.45 | 22.30 |
> | GLOBEM | Qwen3-30B-A3B | 35.63 | 25.75 | 22.30 | 23.91 |
> | GLOBEM | GPT-5-mini | 36.09 | 28.05 | 28.72 | 25.64 |
> ---
> ### Q4: Limited Domains
> - We provide three diverse domains, each with large-scale real-world databases, and a **query-free methodology** for constructing verifiable open-data benchmarks applicable to any domain.
> ---
> Reference:
> 1. Wang, L., Xu, W., Lan, Y., Hu, Z., Lan, Y., Lee, R. K. W., & Lim, E. P. (2023, July). Plan-and-solve prompting: Improving zero-shot chain-of-thought reasoning by large language models. In Proceedings of the 61st annual meeting of the association for computational linguistics.
> 2. Sumers, T., Yao, S., Narasimhan, K. R., & Griffiths, T. L. (2023). Cognitive architectures for language agents. Transactions on Machine Learning Research.
> 3. Wu, Q., Bansal, G., Zhang, J., Wu, Y., Li, B., Zhu, E., ... & Wang, C. (2024, August). Autogen: Enabling next-gen LLM applications via multi-agent conversations. In First conference on language modeling.

---

> > ### Author Rebuttal · Reviewer_XYSz · 2026-04-02
> >
> > The results of mulit-agent is interesting, but why mulit-agent does not work, could you provide more detailed analysis?

---

> > > ### Author Response · Authors · 2026-04-02
> > >
> > > Thank you for your reply! We were also surprised to observe that these agent frameworks underperformed the ReAct baseline, so we conducted an analysis after the additional experiments.
> > >
> > > For the multi-agent setting, we designed two agents: an Explorer agent responsible for tool execution and data exploration, and an Analyst agent without tool access, who reviews the Explorer’s trajectory and provides round-by-round feedback on whether to continue exploration and which aspects to investigate next.
> > >
> > > Through trajectory analysis, we found that the lower performance of this multi-agent framework on DDR-Bench mainly stems from two direct causes: premature self-termination and, to a lesser extent, formatting errors such as malformed JSON outputs that trigger forced termination by the system. Further inspection of the trajectories reveals the following underlying factors.
> > >
> > > - **Complex agent structures increase context difficulty for smaller models**: For smaller models, complex agent workflows substantially increase contextual complexity, making it difficult to follow instructions accurately under long contexts and to produce correctly formatted tool calls. This leads either to three consecutive failed calls that trigger forced termination in the DDR-Bench framework, or to excessive retries that pollute the context. These issues account for approximately 21 per cent of erroneous trajectories, with an average exploration length of 15.6 rounds.
> > >
> > > - **Multi-agent interaction reshapes confidence and encourages premature termination**: The dominant issue is that multi-agent interaction alters the model’s confidence calibration, resulting in early termination of exploration. This phenomenon appears in both small and large models, and is even more pronounced for larger ones. After role separation, the Explorer no longer decides on termination; this is delegated to the Analyst. However, the Analyst has only limited additional information. It does not know the query (since there is no query in DDR-Bench) and lacks awareness of the information density in unexplored database regions, just like Explorer. As a result, it tends to conclude prematurely that exploration is sufficient and instructs termination. Representative cases include:
> > >     - After the Explorer inspects database schemas, computes basic statistics, and performs several high-level analyses as instructed, the Analyst often concludes that the analysis is complete. In reality, these steps correspond only to the classical patterns in the initial exploration phase and do not constitute deep analysis.
> > >     - When the Explorer identifies small amounts of missing data, the Analyst sometimes incorrectly infers that no additional information remains and issues termination instructions.
> > >     - Although we explicitly prompt the Explorer not to self-terminate and to delegate planning decisions to the Analyst, the Explorer tends to signal that it has “fully completed all exploration.” This reflects an instruction-following bias toward satisfying perceived expectations of the Analyst, which in turn leads the Analyst to misjudge the exploration as complete.
> > >
> > > These confidence and coordination issues account for roughly 70 per cent of the failure cases, with an average exploration length of 10.4 rounds.
> > >
> > > **Despite these observations, we believe that improved prompt design, role definitions, and workflow adjustments can surpass the simple ReAct baseline.** Actually, in our experiments, the Plan and Execute framework already achieves better performance. However, optimisation of agent frameworks is inherently open-ended and new corner cases inevitably emerge. Therefore, DDR-Bench focuses on evaluating intrinsic model capability, providing a clean starting point for future evaluation of more sophisticated agent architectures.

---

### Official Review · Reviewer_Swog · 2026-03-13

**Soundness:** 3
**Presentation:** 3
**Significance:** 3
**Originality:** 3
**Overall Recommendation:** 4
**Confidence:** 3

**Summary:**

This paper formalizes Deep Data Research (DDR), an open-ended setting where an agentic LLM is given a database and a minimal toolset (SQL/Python) without predefined questions or interaction limits, and must autonomously explore, decide what to investigate, and self-terminate with a report of mined insights. The authors introduce DDR-Bench, spanning three domains (MIMIC-IV, GLOBEM, 10-K), and propose a checklist-based evaluation: domain facts are extracted (from unstructured components when available) into per-entity checklist items, and an LLM-as-a-checker verifies whether the model’s message-wise insights (Im) or final trajectory-wise report (It) provides sufficient evidence to support each checklist item’s ground-truth answer, yielding accuracy. Beyond final accuracy, the paper reports analyses of test-time scaling, exploration patterns (coverage/entropy), self-termination behavior, and a novelty analysis for “unused” insights via pairwise comparisons.

**Compliance With Llm Reviewing Policy:**

Affirmed.

**Final Justification:**

Thank you to the authors for addressing my questions. I appreciate the clarification, and I will keep my current scores unchanged.

**Key Questions For Authors:**

Please refer to Weakness.

**Limitations:**

No. The paper would benefit from a more explicit limitations section covering: (i) how to interpret final accuracy under unrestricted exploration (capability vs. budgeting/termination), (ii) the gap between estimated inference cost and practical cost under caching/serving optimizations, and (iii) the external validity boundary introduced by the structured+SQL/Python constraint relative to more unstructured, tool-rich real-world settings.

**Strengths And Weaknesses:**

Strengths:
1) Clear problem framing: the paper explicitly separates “executional intelligence” from “investigatory intelligence” and proposes a benchmark designed to isolate the latter in structured data settings.
2) Verifiable evaluation: checklist-based claim verification provides an interpretable alternative to purely subjective rubric/judge scoring, and the per-item structure enables fine-grained error analysis.
3) Rich diagnostics: reporting both Im and It, plus scaling/exploration/termination analyses, makes the benchmark useful for understanding failure modes rather than only producing a leaderboard.
4) Multi-domain coverage: medical, behavioral/psychology, and financial domains help test robustness across different data modalities and reasoning styles (entity-centric but diverse).

Weaknesses:
1) Checklist coverage vs. open-ended insight: by construction, checklist accuracy cannot reward correct and useful insights that fall outside the checklist, and it can produce false negatives under limited exploration budgets (e.g., a capable agent that did not traverse the right evidence path).
2) Comparability of the main table: because exploration is unrestricted and termination is model-controlled, final accuracy mixes “capability” with “budgeting/termination strategy” (how long to explore, how many tokens to spend, how aggressively to emit insights), which can advantage more “persistent/expansive” agents and may be misread as a pure capability ranking.
3) Cross-entity insights are likely under-valued: the task/evaluation is fundamentally per-entity (patient/user/company) with per-entity checklists, so insights that primarily rely on comparing the current entity against other entities or population-level baselines may be systematically ignored unless they map onto entity-specific checklist items. This could underestimate agents that excel at horizontal comparison and cohort-style reasoning.
4) Cost realism: the cost scaling discussion appears to rely on token counts and serving cost assumptions; practical inference cost can be heavily affected by prompt/inference caching and system-level optimizations, which are not discussed and could change cost-efficiency conclusions.
5) Positioning language: emphasizing “a structured database and a generic toolset” may be read as realism/general applicability, but in practice these choices are enabling constraints for verifiable evaluation (and the toolset is minimal rather than broadly “generic”), so the scope/limitations could be stated more explicitly to avoid reader confusion.

---

> ### Author Rebuttal · Authors · 2026-03-29
>
> Thank you very much for the thoughtful review. We hope the following clarifications address the concerns.
>
>
> ### Q1: Checklist Coverage vs Open-Ended Insight
> - For open-ended data analysis, enumerating all possible insights is inherently difficult. This challenge motivated the benchmark as a first step toward studying the problem. In Section 3.2, we conduct a novelty analysis where insights outside the checklist are evaluated via pairwise comparisons to estimate potential false positives. We observe strong consistency between checklist accuracy and novelty.
> - Regarding potential false negatives due to limited exploration, no exploration budget is imposed. The model autonomously decides when to stop exploring. In an investigatory intelligence setting without predefined questions, allocating exploration budget is itself part of the capability being evaluated.
> ---
> ### Q2: Comparability of the Main Table
> - As discussed above, deciding how long to explore is itself part of investigatory intelligence. Appendix C shows that models of similar scale but different generations exhibit substantial differences in exploration willingness.
> - Moreover, longer exploration does not necessarily lead to better results. Please refer to the turn distribution statistics in Appendix I.
> - For controlled comparisons, Section 4.1 presents scaling analyses across three budget dimensions: turns, tokens, and cost, enabling matched-budget evaluation.
> ---
> ### Q3: Cross-Entity Insights Are Likely Under-Valued
> This is an excellent point.
> - Cross-entity tasks were initially considered but are currently challenging for LLMs and difficult to evaluate, as they require macro-level statistical analysis over the entire database. At our database scale, both the number of insights and trajectory lengths may exceed effective context limits. With smaller databases, the task degenerates into a Kaggle-style modelling problem.
> - Cross-entity analysis can nevertheless support entity-level insights. Such behaviour was not restricted but only a few strong models, such as Claude, proactively investigated other entities.
> - We also conducted early experiments on cross-user comparisons in the GLOBEM dataset. However, large inter-user differences made cross-entity insights potentially unreasonable. We therefore adopted single-user cross-time comparisons instead.
> - Above all, we agree that this is a promising direction, and we are actively working on it.
> ---
> ### Q4: Cost Realism
> - No cost simplifications were made. For API-based models, reported cost reflects actual payment, including KV cache hits handled by providers, approximately 70 per cent on average, and typically billed at around one tenth of the standard rate. For open-source models, KV cache hits are tracked, and the same pricing assumption is applied, while per-token prices follow OpenRouter pricing.
> ---
> ### Q5:  Positioning Language
> - Yes, we intentionally simplified the design to provide a clean testbed for evaluating models. We will explicitly discuss this in the limitations section. Thank you for your suggestions!

---

> > ### Author Rebuttal · Reviewer_Swog · 2026-04-02
> >
> > Good work.
> > For Q2, Which figure are you referring to? I notice a clear link between performance across turns and the results in Figure A3.
> > I read the explanation in Section 4.1, but I think the main table still needs more details to make this clearer.

---

> > > ### Author Response · Authors · 2026-04-02
> > >
> > > Thank you for your recognition! **We will revise the main table format to make it clearer.**
> > >
> > > Regarding the supplementary materials in the appendix:
> > > - In Appendix C and Figure A3, we observe that from Qwen2.5 --> Qwen3 --> Qwen3 Next, corresponding to the last column in Figure A3, **newer model versions tend to conduct longer exploration, which leads to better performance**.
> > > - In Appendix I and Figures A10, A11, and A12, we present the distribution of exploration rounds for all models across all scenarios and entities. **Longer exploration does not necessarily lead to better performance**. For example, GLM explores more than DeepSeek across all scenarios, yet its overall metrics are lower than DeepSeek. Similarly, Qwen3-30B-A3B has more exploration rounds than Kimi K2, but its final performance is still significantly worse than Kimi K2. **Longer exploration rounds are undoubtedly an important factor for achieving high scores on DDR-Bench, but our analysis suggests that they are not the only determining factor.** Stronger models can obtain more information through fewer but more detailed exploration steps, thereby improving efficiency. These trends can also be observed in the scaling curves in Section 4.1.

---

### Decision · Program_Chairs · 2026-04-30

**Decision:**

Accept (regular)

**Comment:**

This work introduces the concept of *investigatory intelligence*—the ability of agentic LLMs to autonomously explore data without predefined queries. It proposes DDR-Bench, a checklist-based benchmark built on real-world datasets to evaluate this capability. The paper makes a compelling and timely contribution by shifting evaluation from passive question answering to active, open-ended data exploration, a setting that better reflects emerging agentic use case. Its checklist-based verification offers a practical and scalable alternative to purely subjective evaluation, and the extensive empirical study—including analyses of exploration behavior, scaling, and agent design.

While the paper has some limitations, such as incomplete checklist coverage, potential bias toward entity-level insights, and simplified agent scaffolding, these issues are well acknowledged and substantially addressed in the rebuttal with additional experiments (e.g., multi-agent and planning frameworks) and validation analyses (e.g., checklist construction and agreement), which strengthen confidence in the methodology and conclusions. Importantly, unlike many incremental benchmark papers, this work introduces a new problem formulation with clear conceptual significance and practical relevance. Overall, the novelty of the task, the thoughtful evaluation design, and the strong empirical analysis outweigh the limitations, and I recommend acceptance.